# INFORMATION LATTICE LEARNING

## ABSTRACT

Information Lattice Learning (ILL) is a general framework to learn decomposed representations, called rules, of a signal such as an image or a probability distribution. Each rule is a coarsened signal used to gain some human-interpretable insight into what might govern the nature of the original signal. To summarize the signal, we need several disentangled rules arranged in a hierarchy, formalized by a lattice structure. ILL focuses on explainability and generalizability from "small data", and aims for rules akin to those humans distill from experience (rather than a representation optimized for a specific task like classification). This paper focuses on a mathematical and algorithmic presentation of ILL, then demonstrates how ILL addresses the core question "what makes X an X" or "what makes X different from Y" to create effective, rule-based explanations designed to help human learners understand. The key part here is *what* rather than tasks like generating X or predicting labels X,Y. Typical applications of ILL are presented for artistic and scientific knowledge discovery. These use ILL to learn music theory from scores and chemical laws from molecule data, revealing relationships between domains. We include initial benchmarks and assessments for ILL to demonstrate efficacy.

## 1 INTRODUCTION

With rapid progress in AI, there is an increasing desire for general AI (Goertzel & Pennachin, 2007; Chollet, 2019) and explainable AI (Adadi & Berrada, 2018; Molnar, 2019), which exhibit broad, human-like cognitive capacities. One common pursuit is to move away from "black boxes" designed for specific tasks to achieve broad generalization through strong abstractions made from only a few examples, with neither unlimited priors nor unlimited data ("primitive priors" & "small data" instead). In this pursuit, we present a new, task-nonspecific framework—Information Lattice Learning (ILL)— to learn representations akin to human-distilled rules, e.g., producing much of a standard music theory curriculum as well as new rules in a form directly interpretable by students (shown at the end).

The term *information lattice* was first defined by Shannon (1953), but remains largely conceptual and unexplored. In the context of abstraction and representation learning, we independently develop representation lattices that coincide with Shannon's information lattice when restricted to his context. Instead of inventing a new name, we adopt Shannon's. However, we not only generalize the original definition—an information lattice here is a hierarchical distribution of representations—but we also bring learning into the lattice, yielding the name ILL.

ILL explains a signal (e.g., a probability distribution) by disentangled representations, called *rules*. A rule explains some but not all aspects of the signal, but together the collection of rules aims to capture a large part of the signal. ILL is specially designed to address the core question "what makes X an X" or "what makes X different from Y", emphasizing the *what* rather than generating X or predicting labels X,Y in order to facilitate effective, rule-based explanations designed to help human learners understand. A music AI classifying concertos, or generating one that mimics the masters, does not necessarily produce human insight about what makes a concerto a concerto or the best rules a novice composer might employ to write one. Our focus represents a shift from much representation-learning work (Bengio et al., 2013) that aim to find the best representation for solving a specific task (e.g., classification) rather than strong concern for explainability. Instead of optimizing a task-specific objective function (e.g., classification error), ILL balances more general objectives that favor *fewer*, *simpler* rules for interpretability, and more *essential* rules for effectiveness—all formalized later.

One intuition behind ILL is *to break the whole into simple pieces*, similar to breaking a signal into a Fourier series. Yet, rather than decomposition via projection to orthonormal basis and synthesis

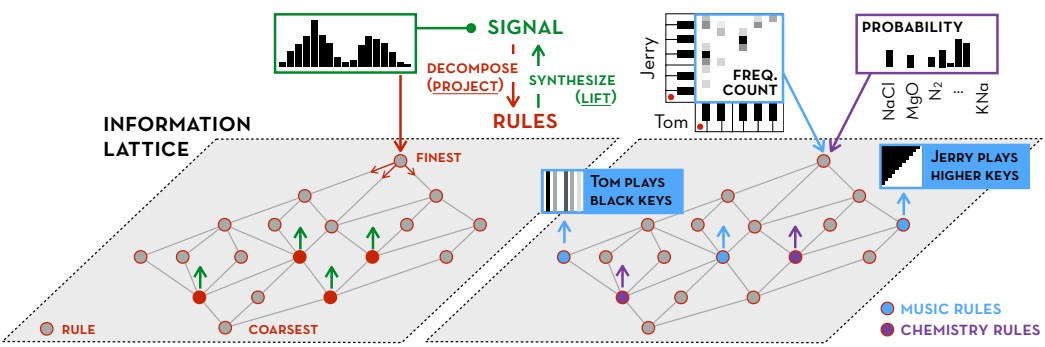

Figure 1: ILL's main idea: decompose the signal into rules that are individually simple but collectively expressive. A lattice is first constructed regardless of the signal (prior-driven), yet the same lattice may be later used to learn rules (data-driven) of signals from different topics, e.g., music and chemistry.

via weighted sum, we decompose a signal in a hierarchical space called a lattice. Another intuition behind ILL is *feature selection*. Yet, rather than features, we use partitions to mimic human concepts and enable structured search in a partition lattice to mimic human learning. The goal is to restore human-like, hierarchical rule abstraction-and-realization through signal decomposition-and-synthesis in a lattice (called projection-and-lifting, Figure 1: left), resulting in more than a sum of parts.

ILL comprises two phases: (a) lattice construction; (b) learning (i.e., searching) in the lattice. This is similar to many machine learning (ML) models comprising (a) function class specification then (b) learning in the function class, e.g., constructing a neural network then learning—finding optimal parameters via back-propagation—in the network. ILL's construction phase is *prior-efficient*: it builds in universal priors that resemble human innate cognition (cf. the Core Knowledge priors (Spelke & Kinzler, 2007)), then grows a lattice of abstractions. The priors can be customized, however, to cater to a particular human learner, or facilitate more exotic knowledge discovery. ILL's learning phase is *data-efficient*: it learns from "small data" encoded by a signal, but searches for rich explanations of the signal via rule learning, wherein abstraction is key to "making small data large". Notably, the construction phase is prior-driven, *not* data-driven—data comes in only at the learning phase. Hence, the same construction may be reused in different learning phases for different data sets or even data on different topics (Figure 1: right). Featuring these two phases, ILL is thus a hybrid model that threads the needle between a full data-driven model and a full prior-driven model, echoing the notion of "starting like a baby; learning like a child" (Hutson, 2018).

ILL is related to many research areas. It draws ideas and approaches from lattice theory, information theory, group theory, and optimization. It shares algorithmic similarity with a range of techniques including MaxEnt, data compression, autoencoders, and compressed sensing, but with a much greater focus on achieving human-like explainability and generalizability. Below, we broadly compares ILL to prominent, related models, leaving more comparisons to the Appendix for most similar ones.

| *Compared to* | *ILL is* |
|---|---|
| deep learning | a "white-box" model balancing human-explainability and task performance |
| Bayesian inference | modeling human reasoning with widely shared, common priors and few, simple rules rather than using probabilistic inference as the driving force |
| tree-like models | structurally more general: a tree (e.g., decision tree or hierarchical clustering) is essentially a linear lattice (called a chain formally) depicting a unidirectional refinement or coarsening process |
| concept lattice in FCA (Ganter & Wille, 2012) | conceptually more general and may include both known and unknown concepts; ILL does not require but discovers domain knowledge (more details in Appendix A) |

We illustrate ILL applications by learning music theory from scores, chemical laws from compounds, and show how ILL's common priors facilitate mutual interpretation between the two subjects. To begin, imagine Tom and Jerry are playing two 12-key pianos simultaneously, one note at a time (Figure 1: right). The frequency of the played two-note chords gives a 2D signal plotted as a $12 \times 12$ grayscale heatmap. Inspecting this heatmap, what might be the underlying rules that govern their co-play? (Check: all grey pixels have a larger "Jerry-coordinate" and project to a black key along the "Tom-axis".) We now elaborate on ILL and use it to distill rules for complex, realistic cases.

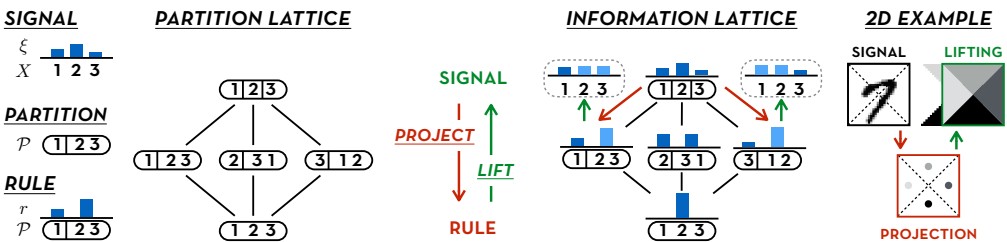

Figure 2: Basic information-lattice terms: signal, partition, rule, lattice, projection and lifting.

## 2  INFORMATION LATTICE: ABSTRACTIONS AND RULES OF A SIGNAL

**Signal.** A *signal* is a function $\xi : X \to \mathbb{R}$. For notational brevity and computational reasons, assume $\xi$ is non-negative and $X \subseteq \mathbb{R}^n$ is finite (not a limitation: see Appendix B). For example, a signal $\xi : \{1, \dots, 6\} \to \mathbb{R}$ being a probability mass function (pmf) of a dice roll, or a signal $\xi : \{0, \dots, 27\}^2 \to \mathbb{R}$ being a $28 \times 28$ grayscale image. We denote the set of all signals on $X$ by $\mathcal{S}_X$.

**Partition / abstraction.** We use a partition $\mathcal{P}$ of a set $X$ to denote an *abstraction* of $X$; we call a cell $C \in \mathcal{P}$ an (abstracted) *concept*. The intuition is simple: a partition of a set renders a "coarse-grained view" of the set, or more precisely, an equivalence relation on the set. In this view, we identify equivalence classes of elements (concepts) instead of individual elements. For example, the partition $\mathcal{P} = \{\{1, 3, 5\}, \{2, 4, 6\}\}$ of the six outcomes of the roll of a die identify two concepts (odd, even).

**Rule / representation.** A *rule of a signal* $\xi : X \to \mathbb{R}$ is a "coarsened" signal $r_\xi : \mathcal{P} \to \mathbb{R}$ defined on a partition $\mathcal{P}$ of $X$ with $r_\xi(C) := \sum_{x \in C} \xi(x)$ for any $C \in \mathcal{P}$. In this paper, a rule of a signal is what we mean by a *representation* of a signal. If the signal is a grayscale image, a rule can be a special type of blurring or downsampling of the image; if the signal is a probability distribution, a rule can be a pmf of the "orbits" of the distribution for lifted inference algorithms (Holtzen et al., 2019; Kersting, 2012). More generally, we define a *rule (regardless of any signal)* over a set $X$ by any signal on any partition of $X$; accordingly, we denote the set of all rules over $X$ by $\mathcal{R}_X := \cup_{\mathcal{P} \in \{\text{all partitions of } X\}} \mathcal{S}_\mathcal{P}$.

**Partition lattice.** Abstractions are hierarchical: one coarse-grained view can be coarser than another. Let the *partition lattice* $(\mathfrak{P}_X, \preceq)$ of a set $X$ be the partially ordered set (poset) containing all partitions of $X$ equipped with the partial order *coarser than* ($\preceq$), or *finer than* ($\succeq$), defined in the standard way. Let $\overline{\mathcal{P}} := \{\{x\} \mid x \in X\}$ and $\underline{\mathcal{P}} := \{X\}$ denote the finest and the coarsest partition, respectively. Per general lattice theory (Davey & Priestley, 2002), $\mathfrak{P}_X$ is a complete lattice: every subset $\mathfrak{P} \subseteq \mathfrak{P}_X$ has a unique supremum $\vee \mathfrak{P}$ and a unique infimum $\wedge \mathfrak{P}$, where $\vee \mathfrak{P}$ is called the *join* of $\mathfrak{P}$ denoting its coarsest common refinement and $\wedge \mathfrak{P}$ is called the *meet* of $\mathfrak{P}$ denoting its finest common coarsening.

**Information lattice.** The *information lattice* $(\mathcal{R}_\xi, \Leftarrow)$ of a signal $\xi : X \to \mathbb{R}$ is the poset of all rules of $\xi$ equipped with the partial order *more general than*: for any two rules $r, r' \in \mathcal{R}_\xi$, we say $r$ is more general than $r'$ (or $r'$ is more specific), denoted $r \Leftarrow r'$, if $\mathrm{domain}(r) \preceq \mathrm{domain}(r')$. Notably, $\mathcal{R}_\xi \subseteq \mathcal{R}_X$ and $\mathcal{R}_\xi$ is isomorphic to the underlying partition lattice via *projection* defined below.

**Projection and lifting.** For any signal $\xi \in \mathcal{S}_X$, we define the *projection* operator $\downarrow^\xi : \mathfrak{P}_X \to \mathcal{R}_\xi$ by letting $\downarrow^\xi (\mathcal{P})$ be the rule of $\xi$ on $\mathcal{P}$. One can check that $\downarrow^\xi : (\mathfrak{P}_X, \preceq) \to (\mathcal{R}_\xi, \Leftarrow)$ is an isomorphism. Conversely, we define the *general lifting* operator $\Uparrow^X : \mathcal{R}_X \to 2^{\mathcal{S}_X}$ by letting $\Uparrow^X (r)$ denote the set of all signals that *satisfy* the rule $r$, i.e., $\Uparrow^X (r) := \{\xi \in \mathcal{S}_X \mid \downarrow^\xi (\mathrm{domain}(r)) = r\} \subseteq \mathcal{S}_X$. To make lifting unique and per Principles of Indifference (Eva, 2019), we introduce a special lifting $\uparrow^X (r)$ to pick the most "uniform" signal in $\Uparrow^X (r)$. Formally, define $\| \cdot \|_q : \mathcal{S}_X \to \mathbb{R}$ by $\|\xi\|_q := (\sum_{x \in X} \xi(x)^q)^{1/q}$. For any $\xi, \xi' \in \mathcal{S}_X$ satisfying $\|\xi\|_1 = \|\xi'\|_1$, we say that $\xi$ is more uniform than $\xi'$ (or $\xi'$ is more deterministic) if $\|\xi\|_2 \leq \|\xi'\|_2$. We define the *(special) lifting* operator $\uparrow^X : \mathcal{R}_X \to \mathcal{S}_X$ by $\uparrow^X (r) := \mathrm{argmin}_{\xi \in \Uparrow^X(r)} \|\xi\|_2$ (can be computed by simply averaging). Notation here follows the convention as to function projections to quotient spaces (Kondor & Trivedi, 2018). Lifting a single rule to the signal domain can be extended in two ways: (a) lift to a finer rule domain $\mathcal{P}$ instead of $X$, i.e., $\Uparrow^\mathcal{P} (r)$ or $\uparrow^\mathcal{P} (r)$; (b) lift more than one rules. Accordingly, we write $\Uparrow := \Uparrow^X$ and $\uparrow := \uparrow^X$ as defaults, write $\mathcal{R} = \downarrow^\xi (\mathfrak{P}) := \{\downarrow^\xi (\mathcal{P}) \mid \mathcal{P} \in \mathfrak{P}\} \subseteq \mathcal{R}_\xi$ to denote a rule set, and write $\Uparrow(\mathcal{R}) := \cap_{r \in \mathcal{R}} \Uparrow(r) = \{\eta \in \mathcal{S}_X \mid \downarrow^\eta (\mathfrak{P}) = \mathcal{R}\}$ and $\uparrow(\mathcal{R}) := \mathrm{argmin}_{\eta \in \Uparrow(\mathcal{R})} \|\eta\|_2$ to denote signals that satisfy all rules in $\mathcal{R}$ (general lifting) and the most uniform one (special lifting), respectively. More computational details on lifting and its intimate relation to join are in Appendix C.

## 3    INFORMATION LATTICE LEARNING (ILL)

We first formalize ILL as a single optimization problem and then solve it practically in two phases. Let $\xi : X \to \mathbb{R}$ be a signal we want to explain. By *explaining*, we mean to search for a rule set $\mathcal{R} = \downarrow^{\xi}(\mathfrak{P}) \subseteq \mathcal{R}_{\xi}$ such that: (a) $\mathcal{R}$ recovers $\xi$ well, or $\mathcal{R}$ is *essential*; (b) $\mathcal{R}$ is *simple*. The main idea agrees with Algorithm Information Theory (Chaitin, 1987; Chater & Vitányi, 2003), but we use an information-lattice based formulation focusing on explainability. We start our formulation below.

We say a rule set $\mathcal{R}$ recovers the signal $\xi$ exactly if $\uparrow(\mathcal{R}) = \xi$. Yet, exact recovery may not always be achieved. The information loss occurs for two reasons: (a) insufficient abstractions, i.e., the join $\vee\mathfrak{P}$ is strictly coarser than $\overline{\mathcal{P}}$; (b) the choice made in favor of uniformity is inappropriate. Instead of pursuing exact recovery, we introduce $\Delta(\uparrow(\mathcal{R}), \xi)$—a distance (e.g., $\ell_p$ distance) or a divergence (e.g., KL divergence) function—to measure the loss, with a smaller $\Delta$ indicating a more essential $\mathcal{R}$.

We say a rule set $\mathcal{R}$ is simpler if it contains fewer and simpler rules. Formally, we want $\mathcal{R}$ *minimal*, i.e., each rule $r \in \mathcal{R}$ is indispensable so as to achieve the same $\uparrow(\mathcal{R})$. Also, we want each rule $r \in \mathcal{R}$ *informationally simple*, measured by smaller Shannon entropy $\mathsf{Ent}(r)$, so $r$ is more deterministic (Falk & Konold, 1997), easier to remember (Pape et al., 2015) and closer to our common-sense definition of a "rule". Notably, the partial order renders a *tradeoff* between the two criteria: $r \Leftarrow r'$ implies $r$ is dispensable in any $\mathcal{R} \supseteq \{r, r'\}$ but on the other hand $\mathsf{Ent}(r) \leq \mathsf{Ent}(r')$, so including more-specific rules makes the rule set small yet each individual rule (informationally) hard.

**The main problem.** The formal definition of an ILL problem is: given a signal $\xi : X \to \mathbb{R}$,

$$\underset{\mathcal{R} \subseteq \mathcal{R}_{\xi}}{\text{minimize}} \quad \Delta(\uparrow(\mathcal{R}), \xi) \qquad \text{subject to} \quad \mathcal{R} \text{ is minimal}; \ \mathsf{Ent}(r) \leq \epsilon \text{ for any } r \in \mathcal{R}. \qquad (1)$$

The search space involves the full information lattice $(\mathcal{R}_{\xi}, \Leftarrow)$, or isomorphically, the full partition lattice $(\mathfrak{P}_X, \preceq)$. Yet, the size of this lattice, i.e., the Bell number $B_{|X|}$, scales faster than exponentially in $|X|$. It is unrealistic to compute all partitions of $X$ (unless $X$ is tiny), let alone the partial order. Besides computational concerns, there are two reasons to avoid the full lattice (but to leave it implicitly in the background): (a) the full lattice has unnecessarily high resolution, comprising many nearly-identical partitions particularly when $X$ is large; (b) considering explainability, not every partition has an easy-to-interpret criterion by which the abstraction is made. As such, Formulation (1) is only conceptual and impractical. Next, we relax it and make it practical via two ILL phases.

### 3.1    PRACTICAL LATTICE CONSTRUCTION: TO START LIKE A BABY (PHASE I)

Information lattice construction plays a role similar to building a function class in ML, sometimes called meta-learning. While its importance is commonly understood, the construction phase in many data-driven models is often treated cursorily—using basic templates and/or ad-hoc priors—leaving most of the computation to the learning phase. In contrast, we put substantial effort into our prior-driven construction phase. Pursuing generality and interpretability, we want *universal*, *simple* priors that are domain-agnostic and close to the innate cognition of a human baby (Marcus, 2018). Here we draw those from Core Knowledge (Spelke & Kinzler, 2007; Chollet, 2019), which include "the (small) natural numbers and elementary arithmetic prior" and "the elementary geometry and topology prior". We then give algorithms to construct abstractions from these priors, and consider such a construction *prior-efficient* if it is interpretable, expressive, and systematic. In the following flowchart, we summarize information lattice construction as generating a partition sublattice.

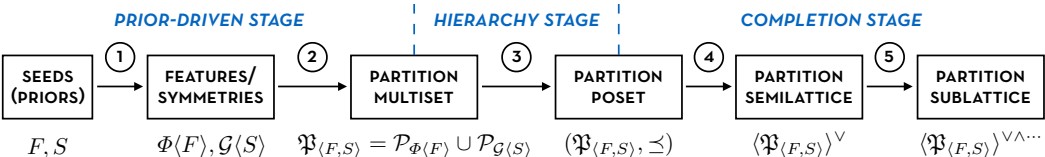

①② **Feature / Symmetry-induced partitions.** Unlike data clustering, our prior-driven partitions are induced from two data-independent sources—features and symmetries. We draw priors—in the form of seed features $F$ and seed transformations $S$—from Core Knowledge as a basis, and then generate a set of partitions $\mathfrak{P}_{\langle F,S \rangle}$ as follows: as an example, for $X = \mathbb{R}^2$:

$$F = \{w_{[1]}, w_{[2]}, w_{[1,2]}, \texttt{sort}, \texttt{argsort}, \texttt{sum}, \texttt{diff}, \texttt{div}_2, \ldots, \texttt{div}_{19}, \texttt{mod}_2, \ldots, \texttt{mod}_{19}\} \qquad (2)$$

$$S = \{\text{horizontal, vertical, diagonal translations}\} \cup \{\text{rotations}\} \cup \{\text{reflections}\} \qquad (3)$$

| $\Phi\langle F\rangle$ : | set of features | generated by $F$ | via function composition |
|---|---|---|---|
| $\mathcal{G}\langle S\rangle$ : | set of subgroups | generated by subsets of $S$ | via subgroup generation |
| $\mathcal{P}_{\Phi\langle F\rangle}$ : | set of partitions | generated by features in $\Phi\langle F\rangle$ | via preimages |
| $\mathcal{P}_{\mathcal{G}\langle S\rangle}$ : | set of partitions | generated by subgroups in $\mathcal{G}\langle S\rangle$ | via orbits |

In (2), $w_I$ denotes coordinate selection (like indexing/slicing in python) and the other functions are defined as in python (div and mod are like in python divmod). Then, $\mathfrak{P}_{\langle F,S\rangle} = \mathcal{P}_{\Phi\langle F\rangle} \cup \mathcal{P}_{\mathcal{G}\langle S\rangle}$.

③ **Partition poset.** We next sort $\mathfrak{P}_{\langle F,S\rangle}$, computationally a multiset, into the poset $(\mathfrak{P}_{\langle S,F\rangle}, \preceq)$. We import algorithmic skeleton from generic poset-sorting algorithms (Caspard et al., 2012; Daskalakis et al., 2011), with an outer routine incrementally adding elements and querying an inner subroutine (an oracle) for pairwise comparison. Yet, our poset is special: its elements are called *tagged partitions* where a tag records the generating source(s) of its tagged partition, e.g., features and/or symmetries. So, we have specially designed both the outer routine ADD_PARTITION and the oracle COMPARE by leveraging (a) *transitivity* (valid for all posets), (b) partition *size* (valid for partitions), and (c) partition *tag* (valid for tagged partitions) to pre-determine or filter relations. We relegate details to Appendix E. The data structures for posets include po_matrix and hasse_diagram, encoding the partial order $\prec$ (ancestors/descendants) and the cover relation $\prec_c$ (parents/children), respectively (Garg, 2015).

④ ⑤ **Partition semi/sublattice.** To complete $(\mathfrak{P}_{\langle F,S\rangle}, \preceq)$ into a lattice, we compute the sublattice (of $\mathfrak{P}_X$) generated by $\mathfrak{P}_{\langle F,S\rangle}$. We follow the idea of *alternating-join-and-meet* completions borrowed from one of the two generic sublattice-completion methods (Bertet & Morvan, 1999). A discussion on our choice and other related methods is in Appendix D. However, we implement join-semilattice completion (meet-semilattice is dual) in our special context of tagged partitions, which echoes what we did in ③ and reuses ADD_PARTITION. The adjustments are (a) changing tags from features and symmetries to join formulae and (b) changing the inner subroutine from pairwise comparison to computing join. We then run a sequence of alternating joins and meets to complete the lattice. For interpretability, one may want to stop early in the completion sequence. While a single join or meet remains simple for human interpretation—often understood as the intersection or union of concepts (e.g., the join of colored items and sized items gives items indexed by color and size)—having alternating joins and meets may hinder comprehension. More details on a single-step join-semilattice-completion, the completion sequence, and tips on early stopping are relegated to Appendix E.

### 3.2 PRACTICAL LATTICE LEARNING: TO LEARN LIKE A CHILD (PHASE II)

Learning in an information lattice means solving the optimization Problem (1), i.e., to search for a minimal subset of simple rules from the information lattice of a signal so as to best explain that signal. Let $\mathfrak{P}_\bullet$ be the sublattice (or semilattice, poset, if early stopped) from the construction phase. Projecting a signal $\xi : X \to \mathbb{R}$ to $\mathfrak{P}_\bullet$ yields the information sublattice $\mathcal{R}_\bullet := \downarrow^\xi(\mathfrak{P}_\bullet) \subseteq \mathcal{R}_\xi$. It is worth reiterating that (a) $\mathfrak{P}_\bullet$ is constructed first and is data-independent; (b) $\xi$ (data) comes after $\mathfrak{P}_\bullet$; (c) $(\mathcal{R}_\bullet, \Leftarrow)$ is isomorphic to $(\mathcal{P}_\bullet, \preceq)$: $\mathcal{R}_\bullet$ retains the partial order (po_matrix and hasse_diagram) and interpretability from $\mathcal{P}_\bullet$. As such, $\mathcal{R}_\bullet$ is what is given at the beginning of the learning phase.

**The main problem (relaxed).** For practicality, we relax Problem (1): instead of the full lattice $\mathcal{R}_\xi$, we restrict the search space to $\mathcal{R}_\bullet$; instead of minimal rule sets, we consider only antichains (whose elements are mutually incomparable), necessary for minimality. This yields:

$$\underset{\mathcal{R} \subseteq \mathcal{R}_\bullet}{\text{minimize}} \ \Delta(\uparrow(\mathcal{R}), \xi) \qquad \text{subject to} \ \mathcal{R} \text{ is an antichain; } \mathsf{Ent}(r) \leq \epsilon \text{ for any } r \in \mathcal{R}. \qquad (4)$$

To solve Problem (4), we adopt a (greedy) idea similar to principal component analysis (PCA): we first search for the most essential rule—which decreases $\Delta$ most—in explaining the signal, then the second most essential rule in explaining the rest of the signal, and so on. Specifically, we start with an empty rule set $\mathcal{R}^{(0)} := \emptyset$, and add rules iteratively. Let $\mathcal{R}^{(k)}$ be the rule set formed by Iteration (Iter) $k$ and $\mathcal{R}^{(k)}_\Leftarrow := \{r \in \mathcal{R}_\bullet \mid r \Leftarrow r' \text{ for some } r' \in \mathcal{R}^{(k)}\}$. Let $\mathcal{R}_{\leq\epsilon} := \{r \in \mathcal{R}_\bullet \mid \mathsf{Ent}(r) \leq \epsilon\}$. Then,

$$(\text{in Iter } k+1) \ \text{ minimize } \ \Delta(\uparrow(\mathcal{R}^{(k)} \cup \{r\}), \xi) \quad \text{subject to} \ r \in \mathcal{R}^{(k)}_{feasible} := \mathcal{R}_{\leq\epsilon} - \mathcal{R}^{(k)}_\Leftarrow. \quad (5)$$

We pre-compute $\mathcal{R}_{\leq\epsilon}$ (instead of the whole $\mathcal{R}_\bullet$) before iterations, which can be done by a breadth-first search (BFS) on $\mathfrak{P}_\bullet$'s hasse_diagram, from bottom (the coarsest) up. As to the monotonicity of $\mathsf{Ent}$ w.r.t. the partial order (cf. the grouping axiom of entropy (Cover & Thomas, 2012)), any BFS branch ends once the entropy exceeds $\epsilon$. (For later use, we save the set $\mathcal{R}_{>\epsilon}$ of ending rules in BFS, i.e., the lower *frontier* of $\mathcal{R}_{>\epsilon}$.) In contrast, $\mathcal{R}^{(k)}_\Leftarrow$ is computed per iteration (by querying $\mathfrak{P}_\bullet$'s po_matrix).

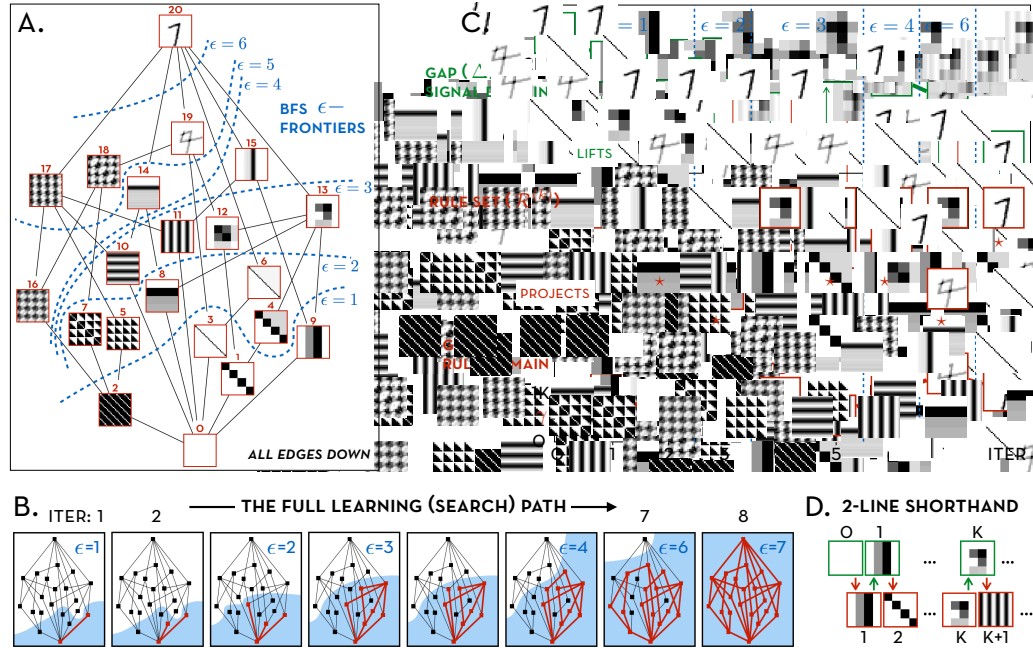

Figure 3: An ILL toy example: (A.) shows the Hasse diagram of an information lattice $\mathcal{R}_\bullet$, demarcated by six levels of $\epsilon$-frontiers. (B.) outlines the full search process along the $\epsilon$-solution path, where the red highlights the activated rules and the blue background shows the marching of the BFS frontier. (C.) shows an ILL output—called a rule trace—where the up and down arrows show the alternating min$\rightleftharpoons$max (lift$\rightleftharpoons$project) optimization and each star marks the best rule set under an $\epsilon$, i.e., a solution to the main relaxed Problem (4). (D.) shows a shorthand of (C.)—called a two-line notation—where the $k$th signal along the signal trace (top line) is recovered by lifting up the first $k$ rules along the rule trace (bottom line) then the $(k+1)$th rule is extracted by projecting down the $k$th signal to the lattice.

**Nested vs. alternating optimization.** Computing $\uparrow(\mathcal{R}^{(k)} \cup \{r\})$ requires solving a minimization, so Problem (5) is a nested optimization: $\operatorname{argmin}_{r \in \mathcal{R}^{(k)}_{feasible}} \Delta(\operatorname{argmin}_{\eta \in \uparrow(\mathcal{R}^{(k)} \cup \{r\})} \|\eta\|_2, \xi)$. One may de-nest the two: instead of comparing rules by lifting them up to the signal domain, we compare them "downstairs" on their own rule domains. So, instead of minimizing (5)'s objective, we

$$\underset{r \in \mathcal{R}_{\le \epsilon} - \mathcal{R}^{(k)}_\rightleftharpoons}{\text{maximize}} \quad \Delta(\downarrow^{\uparrow(\mathcal{R}^{(k)})}(\operatorname{domain}(r)), \downarrow^\xi(\operatorname{domain}(r))) = \Delta(\downarrow^{\uparrow(\mathcal{R}^{(k)})}(\operatorname{domain}(r)), r). \quad (6)$$

The idea is to find the rule domain on which the recovered $\uparrow(\mathcal{R}^{(k)})$ and the target signal $\xi$ exhibit the largest gap. Adding this rule to the rule set maximally closes the gap in (6), and tends to minimize the original objective in (5). Nicely, in (6) the lifting does not involve $r$, so (5) is de-nested, which further iterates into an alternating min$\rightleftharpoons$max (or lift$\rightleftharpoons$project) optimization. Let $r_\star^{(k)}$ be the solution and $\Delta_\star^{(k)}$ be the optimal value in Iter $k$. We update $\mathcal{R}^{(k+1)} := \mathcal{R}^{(k)} \cup \{r_\star^{(k+1)}\} - \{r_\star^{(k+1)}\text{'s } \texttt{descendants}\}$ (so always an antichain), and proceed to the next iteration. Iterations end whenever the feasible set is empty, or may end early if the rule becomes less essential, measured by $|\Delta_\star^{(k+1)} - \Delta_\star^{(k)}| \le \gamma$ in the nested setting, and $\Delta_\star^{(k)} \le \gamma$ in the alternating setting (for some $\gamma$).

**The full learning path & complexity.** We denote a solve process for Problem (6) by $\textsc{Solve}(\epsilon, \gamma)$, or $\textsc{Solve}(\epsilon)$ if $\gamma$ is fixed ahead. To avoid tuning $\epsilon$ manually, we solve an $\epsilon$-path. For $\epsilon_1 < \epsilon_2 < \cdots$, assume $\textsc{Solve}(\epsilon_i)$ takes $K_i$ iterations, we run the following to solve the main relaxed Problem (6):

$$\emptyset = \mathcal{R}^{(0)} \to \textsc{Solve}(\epsilon_1) \to \mathcal{R}^{(K_1)} \to \textsc{Solve}(\epsilon_2) \to \mathcal{R}^{(K_1+K_2)} \to \cdots \quad (7)$$

So, lattice learning boils down to solving a sequence of combinatorial optimizations on the Hasse diagram of a lattice. We walk through the full process (7) via a toy example, starting with a signal $\xi : \{0, \ldots, 27\}^2 \to [0, 1]$ denoting an image of "7" and a toy-sized information lattice of the signal (Figure 3A). The sequence of optimizations (7) proceeds at two paces concurrently: the slower pace is indexed by $\epsilon_i$; the faster pace is indexed by iteration number $k$. As mentioned earlier, the sets $\mathcal{R}_{\le \epsilon_i}$

are pre-computed at the slower pace, with the $(i + 1)$th BFS initialized from $\underline{\mathcal{R}}_{>\epsilon_i}$ (the ending rules in the $i$th BFS). The monotonicity of Ent w.r.t. the partial order assures that these BFSs add up to a single (global) BFS on the entire Hasse diagram, climbing up the lattice from the bottom. This is shown in Figure 3B as the monotonic expansion of the blue region ($\mathcal{R}_{\leq\epsilon}$) explored by BFS. Locally at each iteration along the slower pace, solving Problem (6) is quadratic in the worst case when the feasible set is an antichain (i.e., no order), and linear in the best case when the feasible set is a chain (i.e., totally ordered). Since local BFSs add up to a single BFS with a standard linear complexity, the entire learning phase has a total complexity between linear and quadratic in the number of vertices and edges in the whole Hasse diagram. In general, the denser the diagram is, the lower the complexity is. This is because $\mathcal{R}_{\preceq}^{(k)}$ tends to be large in this case with more descendants activated (i.e., red in Figure 3B), which in turn effectively shrinks the feasible set (i.e., the blue region minus red). For example, unlike the first three iterations in Figure 3B, the 4th iteration ($\epsilon = 3$) activates more than one rules, including the one being extracted as well as all its unexplored descendants. Further, the upper bound is rarely reached. Unlike in this toy example, BFS in practice is often early stopped when $\epsilon$ becomes large, i.e., when later rules become more random. Hence, targeting at more deterministic and disentangled rules only, *not all* vertices and edges are traversed by BFS. In the end of the learning process, for explanatory purposes, we store the entire $\epsilon$-path and the $(\mathcal{R}^{(k)})_{k\geq0}$ sequence instead of just the very last one. This yields a *rule trace* as the standard ILL output, which we present below.

**How to read ILL output.** ILL outputs a rule trace comprising an evolving sequence of rules, rule sets, and recovered signals (Figure 3C). The three sequences are indexed by iteration and by $\epsilon$-path, so the rule set by the last iteration under any $\epsilon$ (starred) is the returned solution to the main Problem (4). We depict a rule by its lifting, since it sketches both the partition and the rule values. Figure 3C gives a full presentation of a rule trace. We also introduce a two-line shorthand (Figure 3D), keeping only the sequence of the recovered signals and that of the rules. A rule trace answers what makes $\xi$ an $\xi$, or what are the best $\epsilon$-simple rules explaining $\xi$. ILL rules are more interpretable than just eyeballing patterns. (a) *The interpretability of the trace* is manifest in its controllability via $\epsilon, \gamma$: smaller $\epsilon$ for simpler rules and larger $\gamma$ for more essential rules. (b) *The interpretability of each rule* is gained from its partition tag—the criteria by which the abstraction is made. A tag may contain several generating sources as different interpretations of the same rule abstraction. Like different proofs of a theorem, a partition tag with multiple sources reveals equivalent characterizations of a structure and thus, more insights of the signal. So, tags are not only computationally beneficial in constructing lattices, but also key to interpretation. We present in-depth analyses on tags in the applications below.

## 4 ILL EXAMPLES

We show typical ILL examples on knowledge discovery in art and science: learning music theory from scores and chemical laws from compounds (while relegating more analyses on handwritten digits to Appendix F). For both, we fix the same priors—$F, S$ in (2)(3)—thus the same lattice. We fix the same parameters: $\epsilon$-path is $0.2 < 3.2 < 6.2 < \cdots$ (tip: a small offset at the beginning, e.g., $0.2$, is used to get nearly-deterministic rules) and $\gamma$ is $20\%$ of the initial signal gap. This fixed setting is used to show generality and for comparison. Yet, the parameters can be fine tuned in practice.

**Music illustration.** Signals are probability distributions of chords encoded as vectors of MIDI keys. Figure 4a) shows such a signal—the frequency distribution of two-note chords extracted from the soprano and bass parts of Bach's C-score chorales (Illiac Software, Inc., 2020)—with the learned rule trace listed below. The first rule is tagged by $\texttt{argsort} \circ w_{[1,2]}$ and has probability all concentrated in one cell whose elements have a larger $y$-coordinate (the black region above the diagonal). So, this is a deterministic rule, echoing the law of "no voice crossing (N.V.C.)", i.e., soprano higher than bass. Checking later rule tags finds laws of voice range (V.R.), diatonic scale (D.S.), and consonant interval (C.I.)—almost all of the main static rules on two-voice counterpoint. Notably, the third rule is tagged by both $\texttt{mod}_{12} \circ w_{[1]}$ and vertical translation invariance. From both feature and symmetry views, this tag identifies the concept of all Cs, all Ds, etc., which is the music concept of pitch class. The feature view explicitly reveals a period of 12 in pitches—the notion of an octave (in defining pitch class); the symmetry view reveals the topology—the manifold where the concepts lie—in this case a 2D torus.

**Chemistry illustration.** Signals are boolean-valued functions indicating the presence of compound formulae encoded as vectors of atomic numbers in a molecule database. Figure 4b) shows a signal attained by collecting two-element compounds from the Materials Project database (Jain et al., 2013) of common compounds. The first rule tagged by $\texttt{div}_{18} \circ w_{[2]}$ is deterministic: Element 2 can never be

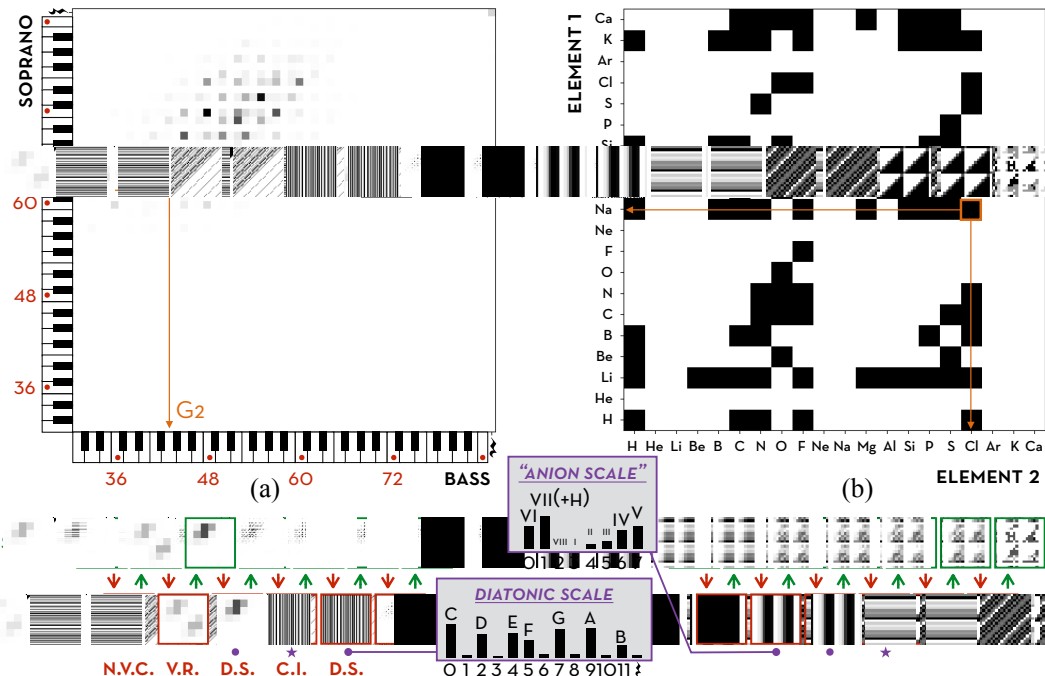

Figure 4: Music and chemical signals (top) with their rule traces learned from ILL (bottom).

Ar, K, Ca. It nicely captures the visual pattern in Figure 4b) (the last three vacant columns) and hints suggestively at some chemistry rules. The second rule tagged by $\text{mod}_8 \circ w_{[2]}$ has peaks at cells tagged by feature values $1, 7, 0, 6$. These cells, for Element 2, are halogens (+H), pnictogens, chalcogens, crystallogens. The third rule shows alkali metals, alkaline earth metals, crystallogens, icosagens are the cells common for Element 1. Next rule shows the common combinations, e.g., alkali metals and halogens. Note that the 2nd, 3rd, 4th rules for chemistry and the 5th, 3rd, 4th rules for music share the same tags, except that $\text{mod}_{12}$ becomes $\text{mod}_8$—period changes from $12$ (a music octave) to $8$ (number of main groups). So, when two chemical elements form a compound, they are like two music notes forming a chord! The music concepts of pitch classes and intervals parallel the chemical concepts of groups and their distances. Although abstractions are shared, rules differ. Instead of a diatonic scale in Bach's chorales, chemistry uses a "cation scale" and an "anion scale". It is interesting that our intention to show ILL's generality (same lattice, parameters for different subjects) also suggests links between art and science by interpreting phenomena (signals) in one subject from the perspective of the other (Bodurow, 2018). Applications that extend the experiment here beyond a clustering model to restore the periodic table (Zhou et al., 2018) and render complex molecules in high dimensions are ongoing, aiming to discover new laws, new interpretations of existing laws, and new materials.

**Real-world deployment & evaluation.** We generalized the music illustration to a real app of an automatic music theorist (Yu et al., 2016; Yu & Varshney, 2017). It specially implements the alternating min⇌max setting into a "student⇌teacher" model: the student is a (music) generator and the teacher is a discriminator. The two form a loop where the teacher guides the student towards a target style through iterative feedback (extracting rules) and exercise (applying rules). This app extends the above music illustration considerably. It considers more music voices, so now signals are in higher dimensions and rules are on more complex chord structure. It considers temporal structure, so now signals include many (un)conditional chord distributions (multi-$n$-grams), yielding both context-free and context-dependent rules, but new challenges too, namely *rare contexts* and *contradictory rules*. ILL's core idea of *abstraction* makes "small data large" thus, rare contexts common (Yu & Varshney, 2017), and a redesigned lifting operator solves contradiction (Yu et al., 2017). Further, parameters like $\epsilon, \gamma$ are made into self-explanatory knobs for users to personalize their learning pace.

We conducted two studies to assess rule-learning capability and interpretability. We present the main results here and detail the procedures in Appendix G. In the first study, we compared ILL-discovered rules with human-codified domain knowledge to see how much known can be reproduced and how much new can be discovered. Trained on just 370 Bach's chorales, our model reproduced in explicit

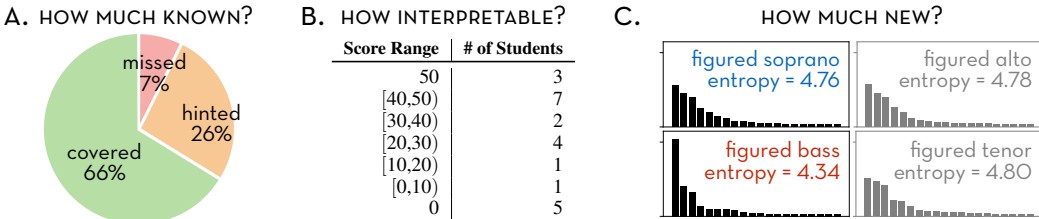

Figure 5: ILL assessments on knowledge discovery tasks.

forms $66\%$ of a standard music theory curriculum (Figure 5A). In the rest, about $26\%$ (e.g., harmonic functions and music forms) was implicitly hinted at by the current $n$-gram based model, modeling only *transitions of abstractions* but not explicitly *abstractions of transitions*—a future direction. In the second study, we ran a human-subject experiment in the form of homework for a music class. The homework asked 23 students to write verbal interpretations of ILL-generated rules rendered as histograms over tagged partitions. Grading was based on a rubric of keywords generated via majority vote in a later discussion among students and teachers. Figure 5B shows that the majority (2/3) of the students who did the homework succeeded (w.r.t. the 30/50 passing grade) in the interpretation task, which in turn shows the interpretability of the AI-produced knowledge itself.

In the first study, our model also discovered new rules that interested our colleagues in the music school. (a) Tritone resolution is crucial in tonal music, yet in Bach's chorales, tritones sometimes do not resolve in typical ways, but consistently transition to other dissonances like a minor seventh. (b) A new notion of "the interval of intervals" was consistently extracted in several rule traces. This "second derivative", like acceleration in mechanics, might suggest a new microscopic chord structure heretofore unconsidered. (c) New symmetry patterns reveal new harmonic foundations. As a parallel concept of harmony traditionally built on *figured bass* (dominant in Bach's chorales confirmed by ILL), ILL reveals "figured soprano" as the next alternative in explaining Bach's music (Figure 5C). Although not the best view for explaining Bach according to ILL and is not included in any standard music theory class, it may be a valuable perspective for music starting deviating from classical. This was confirmed by domain experts (Sokol, 2016), with more details in the end of Appendix G.1.

## 5    DISCUSSION: LIMITATIONS AND CHALLENGES

As a first step, we devise a new representation-learning model intended to be both theoretically sound and intrinsically interpretable. This paper shows typical setups and applications, but ILL is a general framework that admits new designs of its components, e.g., projection-and-lifting or priors. Notably, designing a lattice not only sets the rule-learning capacity but also the "vocabulary" for interpretation which, like the Sapir-Whorf hypothesis for human language, limits how a lattice explains signals. Likewise, priors have pros and cons based on what we seek to explain and to whom (e.g., not all signals are best explained by symmetry, nor can everyone reads symmetry equally well). One solution is to explore multiple lattices while balancing expressiveness and computation—a common practice in picking ML models too. Further, whether a signal is indeed governed by simple rules requires rethinking. Sometimes, no rules exist, then ILL will indicate this and a *case-by-case* study will be needed. Sometimes, rules are insufficient: is music in fact governed by music theory? Theory is better viewed as necessary but not sufficient for good music: great composers need not be great theorists.

Following studies comparing human-codified knowledge and using human-subject experiments for interpretability, more systematic ILL benchmarking and assessment remain challenging and need long-term efforts. Benchmarking is not as easy as for task-specific settings (Chollet, 2019), requiring better comparison schemes or a downstream task. Effective ILL assessments must focus on *new* discoveries and the ability to *assist* people. Instead of a Turing test for machine-generated music, one may (at a meta-level) consider tests between independent and machine-aided compositions, but both are done by humans. Further, ILL may be incorporated with other models, having an ILL version of deep learning or vice versa. For example, using ILL as a pre-processing or post-interpretation module in other models to achieve superior task performance as well as controllability and interpretability. One other possibility may use ILL to analyze attention matrices (as signals) learned from BERT or GPT (Rogers et al., 2020). More future visions are in Appendix H.

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

## A    CONNECTION TO CONCEPT LATTICE

Per our definition, a *concept* refers to a component of an abstraction, or more precisely, is a cell in a partition or an equivalence class under an equivalence relation. This definition is consistent with a *formal concept* defined in formal concept analysis (FCA) (Ganter & Wille, 2012; Ganter et al., 2016; Priss, 2006) as a set of objects (extent) sharing a set of attributes (intent), which can be also treated as objects that are equivalent under the attributes. However, our definition of a concept generalizes that of a formal concept in two ways. First, in our case, a partition or an equivalence relation is not induced from domain-specific attributes through formal logic and formal ontology, but from universal priors drawn from the Core Knowledge (detailed in Section 3.1 in the main paper). Second, specifying a partition considers all of its concepts, whereas specifying a set of formal concepts only considers those with respect to a given *formal context*. As a result, partition lattices in our case generalize concept lattices in FCA, and are not generated, hence not constrained, by domain knowledge such as those encoded in formal ontologies.

Mathematically, let $(\mathfrak{P}_X, \preceq)$ be the partition lattice comprising all partitions of $X$ and $(2^X, \subseteq)$ be the subset lattice comprising all subsets of $X$. Clearly, the power set $2^X$ is the same as $\{C \in \mathcal{P} \mid \mathcal{P} \in \mathfrak{P}_X\}$. That is, the subset lattice is also the lattice comprising all concepts from all partitions of $X$, which can be then called the full concept lattice. So, one can define any concept lattice in FCA as a sublattice of the full concept lattice (cf. Definition 3 in (Ganter et al., 2016)). Yet, such a concept sublattice does not have to include all concepts from a partition, and in many cases, it tends to miss many concepts if they are not known in the existing ontology. We give two examples below to further illustrate the connection between a partition lattice and a concept lattice.

First, consider biological taxonomy. Dogs and cats are two concepts in *species* which is an abstraction containing other concepts such as eagles. Likewise, mammals and birds are two concepts in *class* which is an abstraction containing other concepts such as reptiles and insects; further, animals and plants are two concepts in *kingdom*. In light of hierarchy, as abstractions, species $\succeq$ class $\succeq$ kingdom (in a partition lattice); as concepts, dogs $\subseteq$ mammals $\subseteq$ animals (in a concept lattice). Note that when forming a concept lattice, one may not need to include say, all species. Yet when having species as an abstraction in a partition lattice, this abstraction must contain all species including known species and unknowns, where the latter is usually of more interest for knowledge discovery.

Second, consider music theory. C major triads, C minor triads, and B diminished triads are concepts in an abstraction induced by music octave-shift and permutation invariance. Further, major triads, minor triads, and diminished triads are concepts in another abstraction induced by music octave-shift, permutation, and further transposition invariance. Clearly, for abstractions, the former abstraction is finer than the latter; for concepts, the set of C major triads is a subset (or a special case) of the set of major triads. However, chords that are not defined in traditional music theory but appear as new concepts in a known abstraction (e.g., the two above) may be more interesting, since they may suggest new composition possibilities while still obeying the same music abstraction, in this case the same music symmetry. New concepts from new abstractions may push the composition boundary even further, suggesting new types of chords discovered from e.g., new symmetry (but possibly within a known symmetry family). See the end of Appendix G.1 for more examples from new discoveries.

## B    MORE GENERALIZED FORMALISM FOR INFORMATION LATTICE

The mathematical setting in the main paper is for a non-negative signal on a finite domain. However, this is not a limitation, but purely for notational brevity and computational reasons. First, regarding non-negativity, in many real scenarios, the signal is bounded and its value is only relative. In these cases, one can simply add an offset to the signal to make it non-negative. More generally, we can

consider a signal to be any measurable function $\xi : X \to \mathbb{R}^n$. Then the notions of an abstraction, a concept, a rule, as well as the partial order can be generalized as in Table 1. Hence, the notion of an information lattice is still well-defined in the generalized setting. The essence of the two settings lies in how we formalize an abstraction, whether using a partition or a $\sigma$-algebra. However, the two are not very different from each other: any partition of $X$ generates a $\sigma$-algebra on $X$, and any $\sigma$-algebra on a countable $X$ is uniquely generated by a partition of $X$ (Çınlar, 2011).

Table 1: Formulations in the main paper and their generalizations.

|  | in the main paper | generalized |
|---|---|---|
| signal | $\xi : X \to \mathbb{R}$, $X$ finite | $\xi : X \to \mathbb{R}^n$ measurable |
| abstraction | a partition $\mathcal{P}$ of $X$ | a $\sigma$-algebra $\Sigma$ on $X$ |
| concept | a cell $C \in \mathcal{P}$ | a subset $C \in \Sigma$ |
| rule | $r : \mathcal{P} \to \mathbb{R}, r(C) := \sum_C \xi(C)$ | $r : \Sigma \to \mathbb{R}^n, r(C) := \int_C \xi \, d\mu$ |
| partial order | $\mathcal{P} \preceq \mathcal{P}'$ | $\Sigma \subseteq \Sigma'$ |

Further, the main paper uses the summation functional in defining a rule of a signal, or the projection operator. However, other options are possible, e.g., mean, max, min, or a specially designed functional. The lifting operator can then be redesigned accordingly. In particular, besides always favoring the most uniform signal, the design of the special lifting can have extra freedom in considering other criteria for picking a signal from the general lifting.

## C  MORE INSIGHTS ON THE SPECIAL LIFTING

Consider the special lifting $\uparrow(\mathcal{R})$ for any rule set $\mathcal{R} = \downarrow^\xi(\mathfrak{P})$ of a given signal $\xi$. Computing $\uparrow(\mathcal{R})$ is simple if $\mathcal{R} = \{r\}$ contains only a single rule. In this case, $\uparrow(\mathcal{R})(x) = \uparrow(r)(x) := r(C)/|C|$ for any $x \in C \in \text{domain}(r)$, which requires simply averaging within each cell. However, computing $\uparrow(\mathcal{R})$ becomes much less trivial when $|\mathcal{R}| > 1$. By definition, we need to solve the minimization problem:

$$\uparrow(\mathcal{R}) := \text{argmin}_{r \in \Uparrow(\mathcal{R})} \|r\|_2. \tag{8}$$

Instead of directly throwing the above problem (8) into a generic optimization solver, there is a more efficient approach which also reveals more insights on the special lifting. More specifically, one can check that any multi-rule lifting $\uparrow(\mathcal{R})$ can be computed as a single-rule lifting $\uparrow(r^\star)$ where the single rule $r^\star$ is defined on the join $\vee\mathfrak{P}$ and is computed as follows:

$$r^\star := \text{argmin}_{r \in \Uparrow^{(\vee\mathfrak{P})}(\mathcal{R})} \|\tilde{r}\|_2, \quad \text{with the weighted norm } \|\tilde{r}\|_2 := \sqrt{\sum_C \frac{r(C)^2}{|C|}}. \tag{9}$$

So, instead of lifting $\mathcal{R}$ directly to the signal domain $X$, we lift $\mathcal{R}$ to the join $\vee\mathfrak{P}$ first and then to $X$. Since $|\vee\mathfrak{P}| \leq |X|$, the minimization problem (9) is in a smaller dimension compared to the original problem (8), and thus, can be solved more efficiently. In the minimization problem (9), by definition, $\Uparrow^{(\vee\mathfrak{P})}(\mathcal{R}) := \{r : \vee\mathfrak{P} \to \mathbb{R} \mid \downarrow^r(\mathfrak{P}) = \mathcal{R}\}$. Hence, every rule $r \in \Uparrow^{(\vee\mathfrak{P})}(\mathcal{R})$ can be treated as a single-rule summary of the rule set $\mathcal{R}$, and $r^\star$ is one of them—the one that yields the most uniform signal. Realizing the special lifting $\mathcal{R} \to \uparrow(\mathcal{R})$ as the two-step lifting $\mathcal{R} \to r^\star \to \uparrow(r^\star) = \uparrow(\mathcal{R})$ reveals the following insight: given rules abstracting $\xi$ at different levels (coarser or finer), the best one can hope to *faithfully* explain $\xi$ is at the level of the join. Determining $\xi$ at any level finer than the join would then require additional assumptions other than the rule set itself, such as the preference of uniformity used here. This further explains the two sources of information loss (join and uniformity) discussed in the recovery process of a signal (cf. Section 3 in the main paper). Notably, to determine a signal even at the level of join may be ambiguous, since the general lifting $\Uparrow^{(\vee\mathfrak{P})}(\mathcal{R})$ to the join is not necessarily a singleton. This particularly implies that $r^\star$ as one of the single-rule summaries of $\mathcal{R}$ of $\xi$ is not necessarily a rule of $\xi$, i.e., there is no guarantee that $r^\star = \downarrow^\xi(\vee\mathfrak{P})$. To make it so, we need more rules.

## D    EXISTING WORK ON SUBLATTICE GENERATION

General methods for computing the sublattice $L_B$ of a full lattice $L$ generated by a subset $B \subseteq L$ fall into two basic families, depending on whether the full lattice needs to be computed. The first uses alternating join- and meet-completions, with worse-case complexity $O(2^{|B|})$; the second characterizes the elements of $L$ that belong to the sublattice, with complexity $O(\min(|J(L)|, |M(L)|)^2|L|)$ where $J(L)$ and $M(L)$ denote the number of join-irreducibles and meet-irreducibles, respectively (Bertet & Morvan, 1999). The latter requires computing the full lattice, which is intractable in our case of partition lattices, as $|L| = |\mathfrak{P}_X|$ grows faster than exponentially in $|X|$ whereas $|\mathfrak{P}_{\langle F,S \rangle}|$ is usually smaller than $|X|$. So, we use the first approach and compute alternating join- and meet-completions. The same principle of avoiding computing the full lattice has been applied to the special context of concept lattices (Kauer & Krupka, 2015), yet the technique there still requires the full formal context corresponding to the full concept lattice. Note that sublattice completion is, by definition, computing the smallest sublattice $L_B$ (in a full lattice $L$) containing the input subset $B \subseteq L$, where $L_B$ must inherit the meet and join operations from $L$. It generalizes but is not the same as Dedekind-MacNeille completion (Bertet & Morvan, 1999; MacNeille, 1937; Bertet et al., 1997).

## E    MORE DETAILS ON THE CONSTRUCTION PHASE

This section elaborates on the second half of Section 3.1 in the main paper, presenting more algorithmic details on poset construction and sublattice completion. The core data structures for posets are the so-called *adjacency matrix* and *Hasse diagram*, encoding the partial order $\prec$ and the cover relation $\prec_c$, respectively (Garg, 2015). The former is best for querying *ancestors* and *descendants* of a partition within the lattice; the latter is best for querying *parents* and *children* of a partition. (A more advanced technique includes chain-decomposition, but the two here are sufficient for this paper.) More specifically,

$$\mathcal{P}' \text{ is an ancestor of } \mathcal{P} \iff \mathcal{P} \prec \mathcal{P}'$$

$$\mathcal{P}' \text{ is a parent of } \mathcal{P} \iff \mathcal{P} \prec_c \mathcal{P}' \quad \text{(i.e., } \mathcal{P} \prec \mathcal{P}' \text{ but no } \mathcal{P}'' \text{ satisfies } \mathcal{P} \prec \mathcal{P}'' \prec \mathcal{P}'\text{).}$$

We introduce a few algorithmic notations. Given a partition poset $(\mathfrak{P}, \preceq)$, we use $\mathfrak{P}.\mathtt{po\_matrix}$ and $\mathfrak{P}.\mathtt{hasse\_diagram}$ to denote the adjacency matrix and Hasse diagram of $\mathfrak{P}$, respectively. For any partition $\mathcal{P} \in \mathfrak{P}$, we use $\mathcal{P}.\mathtt{ancestors}$, $\mathcal{P}.\mathtt{descendants}$, $\mathcal{P}.\mathtt{parents}$, and $\mathcal{P}.\mathtt{children}$ to denote the sets of ancestors, descendants, parents, and children of $\mathcal{P}$, respectively. Notably, the two data structures are not only important for the construction phase but for the subsequent learning phase as well. The core subroutine in the construction phase is ADD_PARTITION sketched as Algorithm 1. It is the key unit step in both poset construction and (join-)semilattice completion.

**Poset construction.** This corresponds to Step ③ in the flowchart in Section 3.1 of the main paper. Recall that poset construction refers to the process of sorting a multiset $\mathfrak{P}_{\langle F,S \rangle}$ of tagged partitions into a poset $(\mathfrak{P}_{\langle F,S \rangle}, \preceq)$, where the partition tags are features and symmetries. Naively, if we write an inner subroutine COMPARE$(\mathcal{P}, \mathcal{P}')$—called an *oracle* in the related literature—to compare two partitions, sorting a multiset into a poset amounts to $\binom{N}{2}$ calls of this pairwise comparison where $N$ is the size of the input multiset. So, the common idea shared in almost all poset sorting algorithms is to reduce the number of oracle calls as much as possible. As mentioned in the main paper, considering the additional properties in our case, we leverage (a) *transitivity* (valid for all posets), (b) partition *size* (valid for partitions), and (c) partition *tag* (valid for tagged partitions) to pre-determine or pre-filter relations. In other words, we want to *infer from the context* as many pairwise relations as possible, so that the number of actual pairwise comparisons can be minimized.

More specifically, we start from an empty poset, and call ADD_PARTITION to incrementally add partitions from the input multiset to the poset. As the outer subroutine, ADD_PARTITION leverages transitivity and partition size by maintaining three live data structures, namely `size2partns`, `po_matrix`, and `hasse_diagram`, so as to avoid calling COMPARE whenever possible. Consequently, COMPARE is called only at two places (underlined in Algorithm 1): one for $=$ and one for $\prec$. When called as the inner subroutine, COMPARE$(\mathcal{P}, \mathcal{P}')$ does not always perform an actual computation for pairwise comparison. Instead, it first checks if the tags are informative (e.g., compositions/supergroups imply coarser partitions) and only if not, makes an actual comparison. With the additional information from partition size, an actual comparison can be done in $O(|X|)$ time

---

**Algorithm 1:** ADD_PARTITION $(\mathcal{P}_\tau, \mathfrak{P})$: adds a tagged partition $\mathcal{P}_\tau$ to a partition poset $(\mathfrak{P}, \preceq)$

---

**Input:** a tagged partition $\mathcal{P}_\tau$, where the tag $\tau$ can be a feature/symmetry or a join/meet formula;
      a partition poset $(\mathfrak{P}, \preceq)$, with the following members and hash tables:

- every $\mathcal{P} \in \mathfrak{P}$ is a unique partition (indexed by a unique identifier)
- $\mathfrak{P}$.partn2tags$[\mathcal{P}] := \{\tau \mid \mathcal{P}_\tau = \mathcal{P}\}$ denotes the set of all tags inducing $\mathcal{P}$
- $\mathfrak{P}$.size2partns$[k] := \{\mathcal{P} \mid |\mathcal{P}| = k\}$ denotes the set of all $\mathcal{P} \in \mathfrak{P}$ with size $k$
- $\mathfrak{P}$.po_matrix encodes the partial order $\prec$, best for getting $\mathcal{P}$.ancestors/descendants
- $\mathfrak{P}$.hasse_diagram encodes the cover relation $\prec_c$, best for getting $\mathcal{P}$.parents/children

**Step 1:** determine if $\mathcal{P}_\tau$ is new by $\underline{\text{COMPARE}}(\mathcal{P}, \mathcal{P}_\tau)$ (for $=$) for every
$\mathcal{P} \in \mathfrak{P}$.size2partns$[|\mathcal{P}_\tau|]$

      **if** $\mathcal{P}_\tau \in \mathfrak{P}$.size2partns$[|\mathcal{P}_\tau|]$: update $\mathfrak{P}$.partn2tags$[\mathcal{P}_\tau]$ by adding $\tau$; **return**

      **else**: create a new hash entry $\mathfrak{P}$.partn2tags$[\mathcal{P}_\tau] = \{\tau\}$; **proceed to Step 2**

**Step 2:** add the new partition $\mathcal{P}_\tau$ to $\mathfrak{P}$

  **(2a)** update $\mathfrak{P}$.size2partns$[|\mathcal{P}_\tau|]$ by adding $\mathcal{P}_\tau$

  **(2b)** update $\mathfrak{P}$.po_matrix and $\mathfrak{P}$.hasse_diagram

     – **for** every existing size $k < |\mathcal{P}_\tau|$ sorted in a descending order:
        **for** every $\mathcal{P} \in \mathfrak{P}$.size2partns$[k]$:
          **if** $\mathcal{P}$.parents $\cap \mathcal{P}_\tau$.descendants $\neq \emptyset$: update $\mathfrak{P}$.po_matrix by adding $\mathcal{P} \prec \mathcal{P}_\tau$
          **else**: $\underline{\text{COMPARE}}(\mathcal{P}, \mathcal{P}_\tau)$; update $\mathfrak{P}$.po_matrix and $\mathfrak{P}$.hasse_diagram if $\mathcal{P} \prec \mathcal{P}_\tau$
            (here one can check: it is necessarily the case that $\mathcal{P} \prec_c \mathcal{P}_\tau$)
     – do the above symmetrically **for** every existing size $k > |\mathcal{P}_\tau|$ sorted in an ascending order
     – (note: every $\mathcal{P} \in \mathfrak{P}$.size2partns$[k]$ for $k = |\mathcal{P}_\tau|$ is incomparable with $\mathcal{P}_\tau$)
     – clean cover relation: remove any $\mathcal{P}_* \prec_c \mathcal{P}^*$ from $\mathfrak{P}$.hasse_diagram if $\mathcal{P}_* \prec_c \mathcal{P}_\tau \prec_c \mathcal{P}^*$

---

via a mapping process. More specifically, given two partitions $\mathcal{P}, \mathcal{P}'$, without loss of generality, we assume $|\mathcal{P}| \leq |\mathcal{P}'|$. An actual comparison is made by tentatively creating a mapping $\nu : \mathcal{P}' \to \mathcal{P}$. One can check that such a $\nu$ exists if and only if $\mathcal{P} \preceq \mathcal{P}'$. Hence, if $|\mathcal{P}| = |\mathcal{P}'|$ (resp. $|\mathcal{P}| < |\mathcal{P}'|$), one can determine $=$ (resp. $\prec$) if $\nu$ is created successfully or incomparability otherwise. The mapping complexity is linear in $|X|$, with linear coefficient 1 if mapping succeeds and with linear coefficient $< 1$ if mapping fails. In the worst case (e.g., if all partitions are incomparable), all $\binom{N}{2}$ pairwise comparisons are required. Our algorithm works best when partitions are richly related (i.e., the Hasse diagram is dense), which is indeed the case for our tagged partitions induced from systematically formed features and symmetries.

**Semilattice completion.** This corresponds to Step ④ in the flowchart in Section 3.1 of the main paper. Recall that join-semilattice completion refers to the process of completing a partition poset into a semilattice. We only detail join-semilattice completion, since meet-semilattice completion can be done symmetrically. Formally, we want to compute the join-semilattice of $\mathfrak{P}_X$ generated by the input poset $(\mathfrak{P}_{\langle F, S \rangle}, \preceq)$. We denote the resulting join-semilattice by $\langle \mathfrak{P}_{\langle F, S \rangle} \rangle^\vee$. By definition,

$$\langle \mathfrak{P}_{\langle F, S \rangle} \rangle^\vee := \{ \vee \mathfrak{P} \mid \mathfrak{P} \subseteq \mathfrak{P}_{\langle F, S \rangle} \}.$$

Naively, if computing $\langle \mathfrak{P}_{\langle F, S \rangle} \rangle^\vee$ literally from the above definition, one has to iterate over all subsets of $\mathfrak{P}_{\langle F, S \rangle}$ and compute their joins. This amounts to $2^N$ join computations where $N = |\mathfrak{P}_{\langle F, S \rangle}|$ is the size of the input poset, and moreover, many of the joins are not pairwise. Yet, similar to our earlier poset construction, we may reduce the computations of joins by an incremental method, which also embeds ADD_PARTITION as a subroutine and utilizes partition sizes and tags, but now the tags are join formulae instead of features or symmetries.

More specifically, we start with an empty semilattice $\mathfrak{P}$, and add partitions in $\mathfrak{P}_{\langle F, S \rangle}$ to $\mathfrak{P}$ one by one from smaller-sized to larger-sized (note: the size information is maintained in $\mathfrak{P}_{\langle F, S \rangle}$.size2partns). When a partition $\mathcal{P} \in \mathfrak{P}_{\langle F, S \rangle}$ is to be added, we make a tag named by itself, i.e., let $\mathcal{P}_\tau := \mathcal{P}$ with $\tau := \{\mathcal{P}\}$, and then call ADD_PARTITION$(\mathcal{P}_\tau, \mathfrak{P})$. There are two possibilities here: $\mathcal{P}_\tau$ already exists in $\mathfrak{P}$ (call ends by Step 1) or $\mathcal{P}_\tau$ is new (call ends by Step 2). In the former, we are done with $\mathcal{P}_\tau$.

In the latter, for every $\mathcal{P}' \in \mathfrak{P}\backslash\{\mathcal{P}_\tau\}$, compute the pairwise join $\mathcal{J}(\mathcal{P}') := \vee\{\mathcal{P}_\tau, \mathcal{P}'\}$ and its tags $\mathcal{T}(\mathcal{P}') := \{\tau \cup \tau' \mid \tau' \in \mathfrak{P}.\mathtt{partn2tags}[\mathcal{P}']\}$, and call ADD_PARTITION$(\mathcal{J}(\mathcal{P}')_{\mathcal{T}(\mathcal{P}')}, \mathfrak{P})$. Like COMPARE, computing join can be optimized by leveraging previously computed tags and partial order in the input poset $\mathfrak{P}_{\langle F,S\rangle}$, so as to avoid an actual join computation whenever possible. When inferring from the context is not possible, one can perform an actual join computation $\vee(\mathcal{P}, \mathcal{P}')$ in $O(|X|)$ time. This is done by collecting the unique pairs of cell IDs $(C(x), C'(x))$ for every $x \in X$, where $C(x)$ and $C'(x)$ denote the cell IDs of $x$ in $\mathcal{P}$ and $\mathcal{P}'$, respectively. In the worst case (e.g., if all partitions are incomparable and join-irreducible), the complexity is inevitably $O(2^N)$. However, like in poset construction, our algorithm works best when the partial order structure is rich.

**Practical tips for sublattice completion.** This corresponds to Step ⑤ in the flowchart in Section 3.1 of the main paper. Recall that constructing the sublattice of $\mathfrak{P}_X$ generated by $\mathfrak{P}_{\langle S,F\rangle}$ follows the alternating process: $L_0 := \mathfrak{P}_{\langle S,F\rangle}$, $L_1 := \langle L_0\rangle^\vee$, $L_2 := \langle L_1\rangle^\wedge$, $L_3 := \langle L_2\rangle^\vee$, and so forth, which terminates as soon as $L_{k-1} = L_k$. We denote the end result by $\langle\mathfrak{P}_{\langle S,F\rangle}\rangle^{\vee\wedge\cdots}$, which is the desired sublattice. However, we may want to stop early in the completion sequence, due to concerns from computation, interpretability, expressiveness, as well as their tradeoffs. We suggest a practical tip on deciding where to stop. If the input poset $\mathfrak{P}_{\langle F,S\rangle}$ is small, run alternating joins and meets, or even complete it to the sublattice if affordable. If $\mathfrak{P}_{\langle F,S\rangle}$ is moderate, complete the joins only (as join is closely related to rule lifting, see Appdenix C for more details). If $\mathfrak{P}_{\langle F,S\rangle}$ is large, just use it.

# F    MORE ANALYSES IN THE LEARNING PHASE

This section elaborates on the last paragraph of Section 3.2 in the main paper, presenting more analyses and interpretations on the rule traces elicited from the toy handwritten-digit examples. Yet, as mentioned in the main paper, computer vision is currently not among the typical use cases of ILL. Learning rules of handwritten digits may not be of much independent interest unless for calligraphy. So, the analyses and interpretations here are for illustration purposes only. We refer readers to the Broader Impact section in the main paper for possible future directions on how ILL may be used, together with other ML models, to solve computer vision tasks.

Recall that the main use case of ILL is to *explain* a signal $\xi$, answering what makes $\xi$ an $\xi$. The same toy example illustrating an ILL process is replayed here in Figure 3. The signal $\xi : \{0, \ldots, 27\}^2 \rightarrow [0, 1]$ is a grayscale image of a handwritten "7". In this case, a rule of $\xi$, or the projection of $\xi$ to a partition of $\{0, \ldots, 27\}^2$, can be viewed as gathering "ink" within each partition cell. Accordingly, the (special) lifting can be viewed as redistributing the gathered "ink" (evenly) in each cell. Hence, we term this view the *ink model*. For visual convenience, we depict a rule of a 2D signal by its lifting (i.e., another grayscale image), since with pixels in the same cell colored the same, we can use the lifting to sketch both the partition and the rule values. More precisely, when a lifting represents a rule, it must be viewed in terms of blocks or superpixels; whereas a real lifting (i.e., a signal or a real image) is viewed normally by the regular pixels. To better clarify, all rules in Figure 3 are displayed in red boxes, whereas all liftings are in green ones.

For a simple illustration, we draw a small number of features and symmetries to generate a poset $(\mathfrak{P}_\bullet)$ of 21 partitions. The corresponding part of the information lattice $(\mathcal{R}_\bullet)$ is shown by its Hasse diagram in Figure 3. Further, on top of the Hasse diagram, we demarcate the frontiers of the sublevel sets $(\mathcal{R}_{\leq\epsilon})$ by six blue dashed curves. Note that in this tiny diagram, we have sketched a full range of sublevel sets, yet for large diagrams, sublevel sets are constructed for small $\epsilon$-values only in a single-pass BFS. The right part of Figure 3 illustrates a complete ILL process in the alternating setting, with lift⇌project signified by the green up-arrows and red down-arrows, respectively. During the learning process, ILL tries to minimize the gap in the signal domain (upstairs) through iterative eliminations of the largest gap in the rule domain (downstairs). Eliminating a larger rule gap tends to imply a larger drop in the signal gap, but not necessarily in every iteration, since the special lifting may accidentally recover a better signal if the assumed uniformity is, by chance, present in the signal. The rule set $\mathcal{R}^{(k)}$ formed per iteration is presented in the middle of the right part of Figure 3, which joinly shows the complete rule trace continuously progressing along the $\epsilon$-path.

The rule set in the last iteration under any $\epsilon$ (marked by $\star$ in Figure 3) is the returned solution to the main relaxed Problem (4) in the main paper. This rule set is used to answer what makes $\xi$ an $\xi$. For example, let $r_j$ denote the rule with ID $j$ (here a rule ID is the same as the partition ID, the unique identifier used in Algorithm 1 during the construction phase). Then, among all rules whose entropies

are no larger than $\epsilon = 2$, the third rule set in the trace $\mathcal{R}^{(3)} = \{r_9, r_1, r_{18}\}$ best explains what makes $\xi$ an $\xi$. However, if more complex rules are allowed, say if all rule entropies are now capped by $\epsilon = 6$, $\mathcal{R}^{(7)} = \{r_{13}, r_{15}, r_{19}\}$ is the best.

Recall that we do not just eyeball the rules to get intuitive understandings. Every rule is the projection of the signal to a tagged partition, where the tag, generated in a prior-driven way, explicitly explains the underlying abstraction criteria. For example, $r_{19}$ in Figure 3 comes from a symmetry tag representing a permutation invariance, which visually renders as a reflection invariance. Rules $r_8$ and $r_9$ come from two feature tags $\texttt{div}_7 \circ w_{[1]}$ and $\texttt{div}_7 \circ w_{[2]}$, respectively. These two feature tags represent the continuous and even collapsing in the first and the second coordinate, respectively, which visually render as horizontal and vertical strips in either case. Both rules are later absorbed into $r_{13}$ tagged by $\texttt{div}_7 \circ w_{[1,2]}$, since its rule domain is strictly finer. These rules ($r_8, r_9, r_{13}$) apparently summarize the horizontal and vertical parts of the handwritten "7". Further, the vertical part of the "7" is longer and slants more, so we see more vertically-patterned rules in the rule trace ($r_9, r_{11}, r_{15}$). These rules are obtained from finer and finer abstractions along the horizontal direction, so as to capture more details on the vertical part of that "7" such as its slope. Notably, among these vertically-patterned rules, $r_{11}$ is induced from the symmetry representing a horizontal translation invariance, but it is quickly absorbed into $r_{15}$ whose entropy is not much higher. This transient appearance of $r_{11}$ implies that it plays a less important role in explaining this handwritten "7". In fact, from more experiments, symmetries in general play a less important role in explaining many "7"s. This is, however, not the case in explaining many "8"s, where symmetries occur much more often. For example, consider a symmetry fused from translation and permutation invariances whose fundamental domain is homeomorphic to a Möbius strip. We hypothesize that this topological property might be related to the twisted nature of an "8". For a visual comparison, we present the rule traces learned from a "7" and an "8" below in Figure 6, as well as the visual similarity between a Möbius strip and an "8".

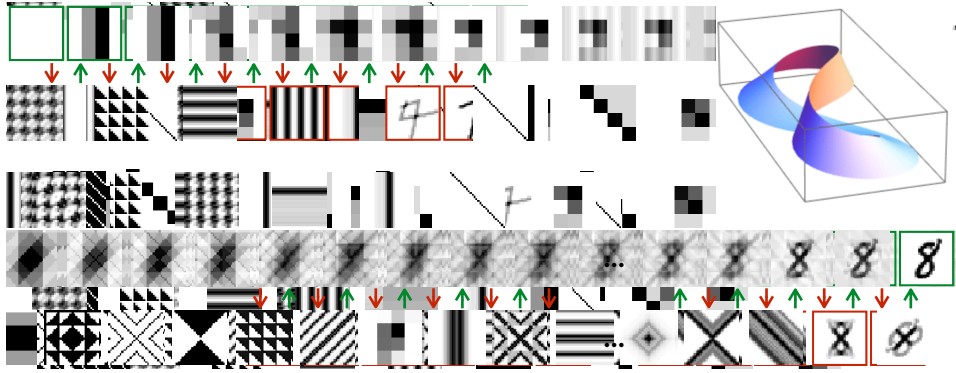

Figure 6: A rule trace of a "7" and that of an "8", together with a 2D Möbius strip.

## G   STUDIES ON ILL-BASED MUSIC APPLICATION

We introduce two tests associated with a real-world application. The first is to assess rule-learning efficacy, where we compare machine-discovered rules to human-codified domain knowledge. The second is to assess human-interpretability, where we use human subject experiments on interpreting machine-generated rules.

The application here is our first step towards building an automatic music theorist and pedagogue, which is to be deployed as an assistant in music research and education. The two tests are our initial effort towards a systematic benchmarking and assessment platform. In the continuing effort of bridging human and machine intelligence, new standards are to be set and commonly agreed upon, so as to reasonably compare machine-codified discoveries with human-codified knowledge, as well as to use human-subject experiments for assessing interpretability. Fully developing assessment protocols is a challenging, long-term endeavor. Here, we use the two tests as starting points, and present results from each. Respectively, the first experiment tests music rule discovery, a basic requirement to be a theorist; the second tests interpretability, a basic requirement to be a pedagogue.

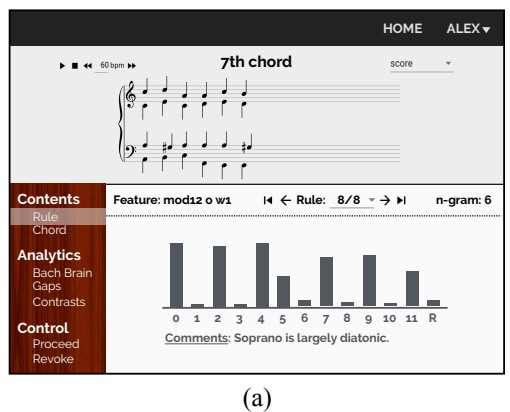 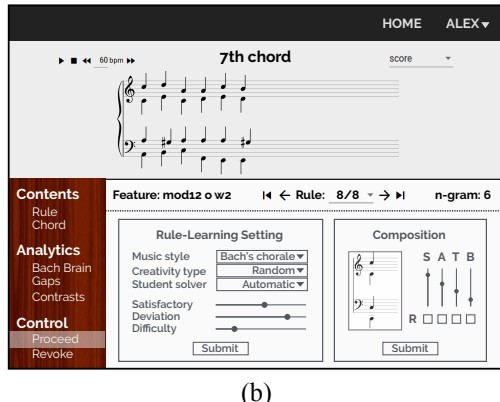

Figure 7: ILL-based music web interface: (a) rule histogram; (b) user control panel.

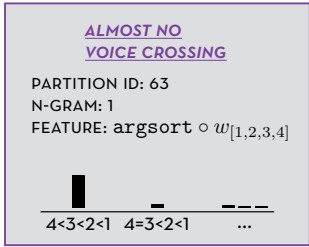 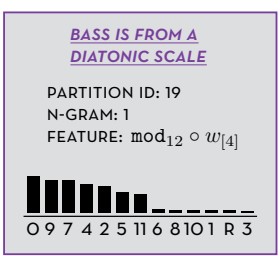 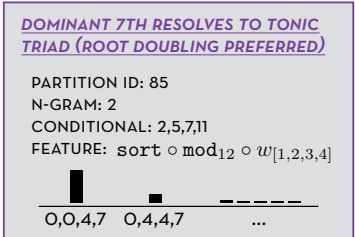

Figure 8: Examples of ILL-generated music rule histograms: first two are context-free rules, the last one is a context-dependent rule.

To conduct the two tests, we first build a user-friendly web application, which is used to better see and control the ILL learning process and results. Figure 7 illustrates the web interface. Users learn music rules—each as a histogram over a tagged partition (i.e., machine-codified music concepts)—and control their learning pace via self-explanatory knobs whose set values are automatically converted to internal parameters (e.g., $\epsilon, \gamma$). One critical music-specific extension to the vanilla ILL presented in the main paper is adding a temporal component, since music is highly contextual. This amounts to considering more than one signal simultaneously, which include various (un)conditional chord distributions (multiple $n$-grams with varying $n$'s and varying conditionals) encoding information of individual chords as well as melodic and harmonic progressions. Accordingly, ILL produces both context-free and context-dependent rules, each of which is indexed by a partition and a conditional under that partition. For example, given the partition that is equivalent to classifying music chords into roman numerals and conditioned on the previous two chords being a $I_4^6$ followed by a V, a rule specifies the probability distribution of the next roman numeral, and in this case reproduces the music rule on Cadential-64. Note that in a context-dependent rule, not only is the query chord abstracted, but also the conditional. This is in contrast with many classical $n$-gram models where no abstraction is present and thus may suffer from the problem of rare contexts, where a conditional occurs very few or even zero times in the training set. However here, the core idea of abstraction makes "small data" large and thus rare contexts common. More examples of context-free and context-dependent rules are illustrated as histograms in Figure 8. These rule histograms are generated from ILL based on 370 of Bach's four-part chorales (in the format of digital sheet music), and are used in the two experiments detailed below.

## G.1 Comparison to Human-Codified Knowledge

We compare rules learned from ILL to a standard undergraduate music theory curriculum. We want to use known laws from music theory as a benchmark to see how ILL-generated rules correspond to human-codified music knowledge. In particular, we want to see what is covered, what is new, and what is different. Yet, the ultimate goal is not just to use known music theory as a ground truth for the purpose of driving ILL to fully reconstruct what we know, but eventually to discover new rules,

to gain new understandings of existing rules, to suggest new composition possibilities, as well as to teach rules in a personalized way.

*A priori* we are aware of three major differences between human-codified music theory and ILL-generated rules. (a) In light of music raw representations (input), laws of music theory are derived from all aspects in sheet music whereas ILL-generated rules are currently derived from only MIDI pitches and their durations. This is because we currently study ILL as a general framework. When a music-specific application is to be developed later, one can include more music raw representations such as letter pitches, meter, measure, beaming, and articulations. (b) In light of rule format (output), laws of music theory and ILL-generated rules have two different styles, with the former being more descriptive and absolute (hard), whereas the latter being more numerical and probabilistic (soft). For instance, a music rule that completely forbids consecutive fifths is reproduced by an ILL-generated rule that assigns a small non-zero probability to the event. Therefore, while it is possible to "translate", with information loss, a (precise) ILL-generated rule to a (verbal) rule in known theory, it may not make sense to "translate" in the opposite direction. Also, it is not a good idea to hardcode known rules as categorical labels in a supervised setting, since music rules are inherently flexible and hardcoding may lead to a rule-based AI that generates somewhat "mechanical" music such as the Illiac Suite (Hiller & Isaacson, 1957). (c) In light of purposes, laws of music theory are more intended for general pedagogical purposes, rather than to reflect the style of a particular data set. For instance, while consecutive fifths are banned in homework and exams, they may be widely used in many pop songs. Even in our data set of Bach's chorales (which are supposed to follow the known rules quite well), we see Bach himself wrote a handful of consecutive perfect intervals. On the contrary, ILL-generated rules are specific to the input data set. We may certainly find some data sets that follow the known rules quite well (e.g., Bach's chorales), but also others that break many known rules and even set their own rules.

Keeping these three differences in mind and by further isolating them from the comparison results, we can reveal the remaining differences that are due to the rule-learning process itself. To come up with the benchmark, we compiled a comprehensive syllabus of laws from music theory taught in our music school's theory review course, which runs through the full series of theory classes at a fast pace. This human-codified music knowledge is organized as a running list of 75 topics and subtopics indexed by lecture number. On the other hand, ILL-generated rules are indexed by partition (ID) and $n$-gram ($n$). The results are summarized below in Table 2, where the colored crosses in the last column indicate topics that are missed by ILL due to different reasons.

Among the total 75 topics in Table 2, we first ignore 7 of them (red crosses) which require music raw representations beyond MIDI pitches and durations (e.g., accents and enharmonic respellings of some augmented sixth chords). ILL covered 45 out of the remaining 68 topics, yielding a coverage of 66%. Among the 23 missed topics, 18 (blue crosses) are related to deeper-level temporal abstractions such as harmonic functions, key areas, and forms. These temporal abstractions may be better modeled as *abstractions of transitions*, which are implicitly captured but not explicitly recovered from our current multi-abstraction multi-$n$-gram language model, modeling only *transitions of abstractions*. The other 5 missed topics (black crosses) are tricky and require ad-hoc encodings, which are not explicitly learnable (but may be implicitly captured to some extent) from our current ILL implementation. Accordingly, the composition of the $30 = 7 + 18 + 5$ uncovered topics suggest three future directions to possibly raise the rule-learning capacity of the current implementation: (a) include more music raw representations; (b) model abstractions of transitions; (c) either make music-specific adjustments when developing music apps or figure out a more expressive and more general framework in the long run. However, remember that the goal here is not to reproduce what we know but to augment it. So, we may certainly stop after enabling abstractions of transitions, which in the best case can yield an improved coverage of 84% (i.e., 93% of the topics from MIDI notes only) which is good enough.

| Lecture | Music Theory | Partition IDs | $n$-gram | |
|---|---|---|---|---|
| 1 | music accents | | | ✗ |
| 2 | pitch | 1-4 | 1 | ✓ |
| 2 | pitch class | 16-19 | 1 | ✓ |
| 2 | interval | 31-36 | 1 | ✓ |

Table 2 (cont.)

| Lecture | Music Theory | Partition IDs | n-gram | |
|---|---|---|---|---|
| 2 | interval class | 97-102 | 1 | ✓ |
| 3 | stepwise melodic motion (counterpoint) | 1-4 | 2 | ✓ |
| 3 | consonant harmonic intervals (counterpoint) | 97-102 | 1 | ✓ |
| 3 | beginning scale degree (counterpoint) | 16-19 | 2 | ✓ |
| 3 | ending scale degree (counterpoint) | 16-19 | 2 | ✓ |
| 3 | beginning interval class (counterpoint) | 97-102 | 2 | ✓ |
| 3 | ending interval class (counterpoint) | 97-102 | 2 | ✓ |
| 3 | parallel perfect intervals (counterpoint) | 97-102 | 2 | ✓ |
| 3 | directed perfect intervals (counterpoint) | | | ✗ |
| 3 | law of recovery (counterpoint) | 1-4 | ≥3 | ✓ |
| 3 | contrapuntal cadence (counterpoint) | 1-4, 97-102 | 2,3 | ✓ |
| 3 | melodic minor ascending line (counterpoint) | | | ✗ |
| 4 | triads and seventh chords | 26-30 | 1 | ✓ |
| 4 | triads and seventh chords: quality | 140-144 | 1 | ✓ |
| 4 | triads and seventh chords: inversion | 113-117 | 1 | ✓ |
| 5 | figured bass | 113-117 | 1,2 | ✓ |
| 5 | roman numerals | 81-85,129-133 | 1 | ✓ |
| 6 | melodic reduction (Schenkerian analysis) | | | ✗ |
| 7 | passing tone (tones of figuration) | 1-4, 134-144 | 3 | ✓ |
| 7 | neighbor tone (tones of figuration) | 1-4, 134-144 | 3 | ✓ |
| 7 | changing tone (tones of figuration) | 1-4, 134-144 | 4 | ✓ |
| 7 | appoggiatura (tones of figuration) | 1-4, 134-144 | 3 | ✓ |
| 7 | escape tone (tones of figuration) | 1-4, 134-144 | 3 | ✓ |
| 7 | suspension (tones of figuration) | 1-4, 134-144 | 3 | ✓ |
| 7 | anticipation (tones of figuration) | 1-4, 134-144 | 3 | ✓ |
| 7 | pedal point (tones of figuration) | 1-4 | ≥ 3 | ✓ |
| 7 | (un)accented (tones of figuration) | | | ✗ |
| 7 | chromaticism (tones of figuration) | | | ✗ |
| 8 | tonic (function) | | | ✗ |
| 8 | dominant (function) | | | ✗ |
| 8 | authentic cadence | 1,4,81-85,129-133 | 2,3 | ✓ |
| 8 | half cadence | 81-85,129-133 | 2,3 | ✓ |
| 9 | voice range (four-part texture) | 1-4 | 1 | ✓ |
| 9 | voice spacing (four-part texture) | 31-41 | 1 | ✓ |
| 9 | voice exchange (four-part texture) | 20-25 | 2 | ✓ |
| 9 | voice crossing (four-part texture) | 53-63 | 1 | ✓ |
| 9 | voice overlapping (four-part texture) | | | ✗ |
| 9 | tendency tone (four-part texture) | 16-19 | 1,2 | ✓ |
| 9 | doubling (four-part texture) | 86-91 | 1 | ✓ |
| 10 | harmonic reduction (second-level analysis) | | | ✗ |
| 11 | expansion chord | | | ✗ |
| 12 | predominant (function) | | | ✗ |
| 13 | phrase model | | | ✗ |
| 14 | pedal or neighbor (six-four chord) | 4,113-117 | 3 | ✓ |
| 14 | passing (six-four chord) | 4,113-117 | 3 | ✓ |
| 14 | arpeggiated (six-four chord) | | | ✗ |
| 14 | cadential (six-four chord) | 85,113-117,133 | 3,4 | ✓ |
| 15 | embedded phrase model | | | ✗ |
| 16 | non-dominant seventh chord (function) | | | ✗ |
| 17 | tonic substitute (submediant chord) | | | ✗ |
| 17 | deceptive cadence (submediant chord) | 81-85,129-133 | 2,3 | ✓ |
| 18 | functional substitute (mediant chord) | | | ✗ |
| 19 | back-relating dominant | 81-85,129-133 | 2,3 | ✓ |
| 20 | period (I) | | | ✗ |
| 21 | period (II) | | | ✗ |
| 22 | period (III) | | | ✗ |

Table 2 (cont.)

| Lecture | Music Theory | Partition IDs | n-gram | |
|---|---|---|---|---|
| 23 | applied chords (I) | 81-85,129-133 | 2,3 | ✓ |
| 24 | applied chords (II) | 81-85,129-133 | 2,3 | ✓ |
| 25 | applied chords (III) | 81-85,129-133 | 2,3 | ✓ |
| 26 | modulation (I) | | | ✗ |
| 27 | modulation (II) | | | ✗ |
| 28 | binary form (I) | | | ✗ |
| 29 | binary form (II) | | | ✗ |
| 30 | modal mixture | | | ✗ |
| 31 | Neapolitan | 81-85,129-133 | 1 | ✓ |
| 32 | Italian sixth chord | 140-144 | 1 | ✓ |
| 32 | French sixth chord | 144 | 1 | ✓ |
| 32 | German sixth chord | | | ✗ |
| 32 | Swiss sixth chord | | | ✗ |
| 33 | ternary form | | | ✗ |
| 34 | sonata form | | | ✗ |

Table 2: Comparison of ILL-generated rules to human-codified laws of music theory taught in standard undergraduate music theory courses. Checks (45) in the last column denote topics recovered by ILL. Red crosses (7) denote topics not recoverable from our music raw representations; blue crosses (18) denote topics not recoverable from our $n$-gram transitions of abstractions (partitions); black crosses (5) denote topics not recoverable from the constructed lattice of abstractions.

From another source of music theory considering music symmetries (Tymoczko, 2010), we compare ILL-generated rules with a set of commonly used music operations, known as the OPTIC operations, namely octave shifts (O), permutations (P), transpositions (T), inversions (I), and cardinality changes (C). The results are summarized in Table 3, which shows that ILL covers the major four types of operations (OPTI). The music C operation is not recovered since it is not a transformation in the mathematical sense. Notations: $t_v$ denotes a translation by the translation vector $v$, i.e., $t_v(x) := x+v$; $r_A$ denotes a rotation (can be proper or improper) by the rotation matrix $A$, i.e., $r_A(x) := Ax$. As a special type of rotation matrices, $P^{(\cdots)}$ denotes a permutation matrix where the superscript is the cycle notation of a permutation. Note that ILL, as a general framework, considers a much larger universe of generic symmetries (from Core Knowledge) beyond those already considered in music. Therefore, ILL can not only study existing music symmetries, but also suggest new symmetries to be exploited in new music styles as well as possible music interpretations of symmetries discovered in other fields like chemistry as described in the main paper.

Table 3: Comparison of ILL's symmetry-induced rule abstractions to the music OPTIC operations.

| Operation | Music Description | Subgroup | |
|---|---|---|---|
| Octave shift | "Move any note into a new octave." | $\langle\{t_{12e_1}, t_{12e_2}, t_{12e_3}, t_{12e_4}\}\rangle$ | ✓ |
| Permutation | "Reorder the object, changing which voice is assigned to which note." | $\langle\{r_{P^{(1,2)}}, r_{P^{(2,3)}}, r_{P^{(3,4)}}\}\rangle$ | ✓ |
| Transposition | "Transpose the object, moving all of its notes in the same direction by the same amount." | $\langle\{t_1\}\rangle$ | ✓ |
| Inversion | "Invert the object by turning it 'upside down'." | $\langle\{r_{-I}\}\rangle$ | ✓ |
| Cardinality Change | "Add a new voice duplicating one of the notes in the object." | | ✗ |

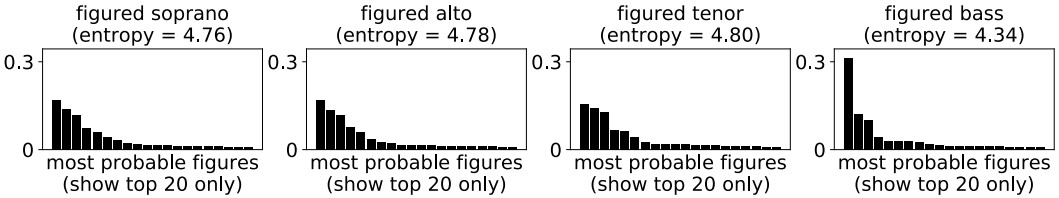

Figure 9: Rules on figured soprano, figured alto, figured tenor, and figured bass. Each histogram shows only the top 20 most probable figures (for brevity), while the entropy is computed from the entire rule. Among the four cases, ILL shows that figured bass (with the smallest entropy) is most effective in explaining Bach's chorales, echoing its importance as a main topic in any standard music theory curriculum. ILL further reveals figured soprano as the next most effective.

Lastly, we mention a few new rules discovered by ILL that are interesting to our colleagues in the School of Music. First, tritone resolution plays an important role in tonal music and appears as an epitome in many more general harmonic resolutions. Yet, in Bach's chorales, tritones are sometimes not resolved in a typical way but consistently transition to another dissonance like the minor seventh, which behaves like a harmonic version of an escape tone or changing tone. Second, a new notion of "the interval of intervals" has been consistently extracted in several ILL-generated rule traces. Such a "second derivative", like acceleration in mechanics, might suggest a new microscopic chord structure to consider. Third, new symmetry patterns reveal new possible foundations for building chords, and thus new composition possibilities. For example, as a parallel concept of harmony traditionally built on *figured bass* (which is indeed the dominant pattern in Bach's chorales confirmed by ILL), ILL reveals "figured soprano" as the next alternative in explaining Bach's music (Figure 9). Clearly, figured soprano is not the best view for explaining Bach according to ILL and is indeed not included in any standard music theory class, yet it may be a more efficient perspective to view other types of music (e.g., in some Jazz improvisations). This vision coincides with comments made by Casey Sokol (Sokol, 2016), a music professor at York University, which we quote below: "The idea of Figured Soprano is simply a way of taking this thinking from the top-down and bringing it into greater prominence as a creative gesture. So these exercises are not anything new in their ideation, but they can bring many new ideas, chord progressions and much else. It's a somewhat neglected area of harmonic study and it's a lot of fun to play with."

### G.2 Human Subject Experiment for Assessing Interpretability

Our second experiment is a human subject experiment, where we collect and assess human-generated verbal interpretations of ILL-generated music rules rendered as sophisticated symbolic and numeric objects. Our goal is to use the results here to reveal both the possibilities and challenges in such a process of decoding expressive messages from AI sources. We treat this as a first step towards (a) a better design of AI representations that are human-interpretable and (b) a general methodology to evaluate interpretability of AI-discovered knowledge representations. In this experiment, we want to test to what degree our ILL-generated rules are interpretable. Our subject pool includes people who have entry-level math and music theory knowledge. So, by *interpretability*, we mean interpretable to them. The whole experimental procedure divides into two stages. At the first stage, we collect human interpretations of ILL-generated rules. At the second stage, we assess the collected interpretations to further evaluate the interpretability of AI-produced knowledge.

**Collect Human Interpretations.** The experiment was conducted in the form of a two-week written homework assignment for 23 students. Students came from the CS+Music degree program recently launched in our university. Entry-level knowledge of computer science, related math, and music theory is assumed from every student. However, all students are new to our AI system, and none have read any ILL-generated rules before. The homework contained three parts. Part I provided detailed instructions on the format of the rules as exemplified in Figure 8, including both feature-related and probability-related instructions (symmetries were excluded from the tags since group theory is an unfamiliar subject to these students). More specifically, we provided verbal definition, mathematical representation, and typical examples for each of the following terms: chord, window (for coordinate selection), seed feature, feature, rule, $n$-gram, histogram, data set. A faithful understanding of these eight terms was the only prerequisite to complete the homework. The estimated reading time of the

instructions was about an hour. Once this self-training part was completed, the students were ready to go to the second and third parts—the main body of the homework. Part II contained eleven 1-gram rules—a histogram specified by window and seed feature(s); Part III contained fourteen 2-gram rules—a histogram now specified by window, seed feature(s), and a conditional. The students were asked to freely write what they saw in each of the histograms guided by the following two questions. (a) Does the histogram agree or disagree with any of the music concepts/rules you know (write down the music concepts/rules in music-theoretic terms)? (b) Does the histogram suggest something new (i.e., neither an agreement nor a disagreement, with no clear connection to any known knowledge)? Answers to each of the 25 rules came in the form of text, containing word descriptions that "decode" the histogram—a symbolic and pictorial encoding. Students were explicitly instructed that writing out a description that was basically a literal repetition of the histogram (e.g., taking a modulo 12 of a chord results in a $91.2\%$ chance of being $0, 0, 4, 7$) is not acceptable: they must reveal the music behind the math. In fact, we made it clear to the students that we only want qualitative descriptions. Students were specifically told (in the instructions) to only pay attention to the relative values of the probabilities whose exact numbers are unimportant (e.g., what are most likely, what are more likely, what are almost impossible). This homework was due in two weeks. During the two-week period, we asked the students to complete it independently, with no group work or office hours.

**Assess Human Interpretations.** The homework was designed in a way such that every rule histogram encoded at least one music concept/rule consistent with standard music theory. In addition, every histogram contained either one additional known music rule or something strange that either conflicted with a known rule or represented something new. We assigned two points per rule. Further, we made an initial rubric containing the (authoritative) music keywords used to describe every rule histogram. Because students' answers arrived in the form of qualitative text, to ensure credibility and fairness of the initial rubric, we held a discussion session at a regular lecture time (80 minutes) with all students as well as the teaching staff. During the discussion session, we went over all 25 rules one by one. For each, we first announced the keywords in the initial rubric and explained to the students that these keywords would later be used to grade their homework. However, in the discussion session, every student was encouraged to object to any of our announced keywords and/or to propose new keywords accompanied with a convincing explanation. New/modified keywords that were commonly agreed upon were added/updated to the initial rubric. By the end of discussion session, we compiled a more inclusive rubric containing broadly accepted keywords. This rubric-generating process was transparent to all the students. In the final step, we manually graded every student's answer sheet against keywords in the rubric and computed their scores. A summary of the students' performances is presented in Table 4. Except for cases where the student did not do the homework, a major source of score deduction was from misunderstanding the $n$-gram (e.g., the probability of the current chord conditioned on the previous chord was mistakenly interpreted as the probability of the previous chord conditioned on the current one). This may be largely due to unfamiliarity with the $n$-gram models for new CS+Music students. Nevertheless, the majority of the students who did the homework (2/3) succeeded (with respect to the 30/50 passing grade) in interpreting the rules generated from ILL, which in turn provides evidence on the interpretability of the AI-produced knowledge itself.

Table 4: Students' final scores.

| Score Range | # of Students |
|---:|:---:|
| 50 | 3 |
| [40,50) | 7 |
| [30,40) | 2 |
| [20,30) | 4 |
| [10,20) | 1 |
| [0,10) | 1 |
| 0 | 5 |

## H   CONCLUSION AND BROADER IMPACTS

Model transparency and interpretability are important for trustworthy AI, especially when interacting directly with people such as scientists, artists, and even multidisciplinary researchers bridging *the Two*

*Cultures* (Snow, 1959) (e.g., like music and chemistry). The core philosophy underlying ILL arises from a human-centered standpoint and our long-term pursuit of "getting humanity back into artificial intelligence". We strive to develop human-like artificial intelligence, which in turn may help advance human intelligence—a goal at the intersection of AGI (artificial general intelligence (Goertzel & Pennachin, 2007)), XAI (explainable artificial intelligence (Adadi & Berrada, 2018)), and "AI as augmented intelligence" (Jordan, 2019).

As such, the focus of interpretability in this line of research is not just the end result of the model, but the entire learning process. This emphasis on *process* is not only manifest in this paper (e.g., two-phase learning that "starts like a baby and learns like a child" with a full rule trace as output), but also in ongoing ILL-driven real-world projects aimed for beneficent societal impact. To name a few: (a) ILL-aided scientific research to accelerate new discoveries, as in biology (Yu et al., 2019); (b) ILL-aided artistic creation to enable new ways and new dimensions in one's creative and/or collaborative experience (art as a process is about more than the work itself); (c) ILL-aided personalized education. Discovered scientific knowledge, artistic expression, and educational curricula, may have a dual use character (Kaiser & Moreno, 2012). Nevertheless, making the discovery of abstract knowledge easier may lead to abstraction traps (Selbst et al., 2019) in deploying such learned knowledge in engineering design or policy making.

Evaluation for ILL and similar technologies should have a humanist perspective, whether comparing to human-codified knowledge or with human subject experiments to assess interpretability. Moreover, evaluations for scientific discovery, artistic creativity, and personalized education should not only focus on model performance, but also on the human-centered criteria of how effectively they aid people in achieving their goals. Illuminating rules associated with practice not only helps human students be better rule-followers, but more creative rule-breakers and rule-makers. Instead of a Turing test for machine-generated music, one might more productively conduct artistic evaluation at a meta-level between human-written music constructed with and without assistance from ILL.

Regarding biases in data, because ILL works in the "small data" regime makes it easier to curate data to avoid representation biases (Suresh & Guttag, 2019). Manually curating 370 music compositions is possible, but manually curating a billion is not.

ILL can be treated as a complement to many existing AI models, with a special focus on model transparency and explainability. Extensions to ILL could enable it to better cooperate with other models, e.g., as a pre-processing or a post-interpretation tool to achieve superior task performance as well as controllability and interpretability. One such possibility could leverage ILL to analyze the attention matrices (as signals) learned from a Transformer-based NLP model like BERT or GPT (Rogers et al., 2020).

