# OpenReview forum: "Information Lattice Learning"
_ICLR.cc/2021/Conference — Reject_

### Official Review · AnonReviewer1 · 2020-10-26
**A reasonable approach but leaves a lot to be desired.**

**Rating:** 6
**Confidence:** 3

**Review:**

The authors perform a descriptive analysis of data by attempting to identify elements in the partial ordering of all partitions on the data which admit a compact definition. Compact definitions are those that are formed by composition of a small number of predefined (prior) set of mathematical operations. Projection and lifting operations are defined to relate descriptions of partition cells to one another through rules. The quality of a description is measured by the divergence between the data and the (special) lifting of the rule set, under the constraint that rules satisfy an upper bound on their entropy.

The approach is general, but due to the intractable size of the information lattice (set of all partitions), simplifications are necessary to produce tractable algorithms. Thus, the authors rely on predefined sets of mathematical operations. This set serves as the defacto language of the summarizations that result. The trouble is that, while the authors describe their method as interpretable, this reviewer finds it very difficult to interpret the summarizations even on the toy problems presented. Moreover, one might refute this difficulty by requiring the user to specify the terms in which they would like to describe the data. Even if a user were capable of doing this, many concepts humans might use are prohibitively difficult to define mathematically, both as a functions and compositions.

General human level summarization of data is a very important task in ML/AI. In the opinion of this reviewer, the community has not adequately solved this problem. The submitted work attempts to move the line forward, but faces a fundamental challenge. We may summarize a set of data by appealing to various groupings of said data (i.e. those that represent fundamental concepts), but we still face the problem of summarizing those groupings. We have only kicked the can down the road so to speak.

The paper is very dense in terminology, which is sometimes conflicting. The authors clearly state that a signal is a function from data to the reals, but then use the same term to describe images. The authors appear to use both 'X' and 'signal' to describe input data. The very heavy use of appendices appears to be a work-around to stuff a great deal of content into the 8-10 page limitation. It makes the paper feel disconnected.

---

> ### Author Response · Authors · 2020-11-25
> **Response to R1 (regarding the fundamental challenge)**
>
> We agree with R1 that formalizing many human concepts is very difficult. In fact, this is one of the main motivations of this work, where we develop a model in assisting humans to reduce such a difficulty. This also matches our focus on human-interpretable representation learning (e.g., knowledge discovery) rather than a task-specific performance score like an error rate. As R1 mentioned that the community is still on its way towards solving that fundamental problem, we do not claim that this work solves it. Instead, we treat our proposed framework as a customizable (e.g., through priors) tool to help people take a step closer in achieving general human-level summarization of data.
>
> To do so, we follow the nativism assumption that more advanced human concepts may be formed later on from a conceptual base—the primitive cognitions in a “babymind”. So, instead of all human concepts, we mathematize only the primitive ones, e.g., those in Core Knowledge. And through our learning framework and knowledge discovery experiments, we check how many existing (or new) concepts can be recovered (or discovered) later on. The results will show not only how the model performs, but also identify which part of the “babymind” is active when summarizing signals from different topic domains. As an example in the paper, observing similar rule traces in music and chemistry not only gives us a mutual interpretation between the two topics, but also indicates a common part of the “babymind” is active in expressing certain music and chemistry phenomena. This can potentially serve as a feedback message to help refine the selections of priors as well as the ways of combining them.
>
> Further, we strongly affirm R1's comment on “data groupings” which indeed echoes the core idea in our formulation of a rule:
>
> * defining rules as patterns on partitions (i.e., “appealing to various groupings of said data”) and
> * deriving partitions from priors (i.e., “the problem of summarizing those groupings”).
>
> For the latter in particular, our feature-, symmetry-, and order-induced partitions via functional, group-theoretic, and lattice-theoretic means, respectively, provide mechanistic interpretations of the groupings. And these mechanistic interpretations are recorded in explicit forms as the so-called partition tags.
>
> In short, our goal in this paper is to propose our framework not as a perfect answer to that fundamental question but a tool to help us address that question. And as a starting point of this journey, we start from some specific topic domains such as music and chemistry.
>
> We thank the reviewer for these comments and believe that attending to them has substantially improved the manuscript.

---

> ### Author Response · Authors · 2020-11-25
> **Response to R1 (regarding interpretability & terminology)**
>
> We are grateful that all the central ideas from our proposed framework regarding goals and motivations as well as algorithmic developments were clearly understood.
>
> **Regarding interpretability.** We apologize that interpretability could have been more clearly presented, and have revised the paper accordingly. We assess interpretability more formally by a separate human-subject study in the form of homework assigned to a class (the 2nd last paragraph in Section 4 and Figure 5B). Unlike for common-sense recognition tasks, interpretability for domain knowledge is not “immediate”, requiring a bit of preparation and analysis. For example, instead of immediately putting their hands on writing the homework, students need to first read an instruction on the format and specification of the homework. In fact, this interpretation test involves both reading and writing the homework independently.
>
> To get an intuition from the toy example, it may be helpful for us to first re-explain that hypothetical but simplest example in Figure 1 about Tom and Jerry’s co-play. The top bigger heatmap is the signal, which maps a pixel location (i.e., a 2-note chord) to its pixel intensity (i.e., the frequency count of the chord). The two smaller heatmaps in the two blue tags below denote two rules. In this simple case, one may eyeball the heatmaps and get their meanings, which we point out below.
>
> * Rule 1: “Jerry always plays a higher key than Tom”.
>
>  This is seen by the fact that all grey pixels are located above the diagonal, i.e., all chords played so far have a higher “Jerry-coordinate”.
>
> * Rule 2: “Tom always plays black keys”.
>
>  This is seen by the fact that if we project all grey pixels to the “Tom-axis”, they all land on black keys.
>
> Besides getting a sense of how 2D rules and signals are visualized in this paper, we also use this starting example to sketch the idea of how coarsening a signal may disclose deterministic patterns that correspond to human-distilled rules. For later examples, instead of eyeballing and guessing, the partition tags are there to tell precisely how the signal is coarsened, e.g., through preimages or orbits or join/meet. We have revised the paper to make both the abstraction generating process (Paragraph 2, Section 3.1) and how to read a rule trace (last paragraph, Section 3.2) clearly stated, so they will help understand the following experimental results.
>
> **Regarding terminology.** We thank R1 for highlighting this confusion. We regret that things did not feel unapproachable. Our sole purpose with the introduced terminology was to formalize the main questions and notions in AGI and XAI. To clarify possible confusions, a *signal* $\xi: X \to \mathbb{R}$ is generally defined and can flexibly refer to different things depending on what “data” means in different contexts. For example, a signal can represent *a single image* in one context, or *a probability distribution of images* in another. In the former, a signal maps a grid point in a 2D grid (“canvas”) to a pixel intensity (“color”) hence $X$ is a “canvas”. In the latter, a signal maps a whole image to a probability hence $X$ is a set of images.
>
> **Regarding appendix.** We apologize and agree that the initial manuscript felt disconnected because of the heavy use of appendices. As our initial attempt to best introduce a new framework, we adopted a top-down view, and prioritized to first present the global picture of the motivation, the problem, the approach, as well as the empirical illustrations as an entirety. We attempted to make the main text as a self-contained story and make the appendix as only supportive but not additional content of that story. We agree that the original submission contained some places that feel disconnected, so we have taken a full pass of revision to smooth out possible gaps and welcome more feedback on this revision.

---

### Official Review · AnonReviewer3 · 2020-10-28
**Interesting new direction**

**Rating:** 7
**Confidence:** 3

**Review:**

This paper proposes a novel learning framework called information lattice learning. It is formulated as an optimization problem that finds decomposed hierarchical representations that are efficient in explaining data using a two-phased approach. ILL generalizes Shannon's information lattice and authors demonstrate ILL can be applied to learning music theory from scores and chemical laws from molecular data. This paper is proposing a new research direction and I believe it is worth to be presented. One concern I have is the complexity and scalability of the proposed algorithm.

Authors emphasize "small data", but I don't see why the proposed approach cannot be applied to "large data". In page 15, authors mention the worst case complexity of O(2^N). Does it mean the proposed approach works only for "simple" examples such as discovering music theory and chemical laws considered in this paper? Can authors elaborate more on the complexity and the scalability issues of their algorithm? Did authors only consider "small data" regime due to the scalability problem?

The definition of signal seems very general and it can even include pmf's. How can we enforce restrictions on signals such as probability simplex?

Can authors comment on how to make a deep learning version of the proposed framework? Say, hierarchical info GAN, hierarchical VAE, etc.?

It would be interesting to compare their work with existing unsupervised deep learning algorithms that attempt to find disentangled representations.

---

> ### Author Response · Authors · 2020-11-25
> **Response to R3**
>
> We are pleased to hear that R3 thinks our proposed learning framework is novel and deserves publication, and wish to thank the reviewer for this endorsement.
>
> **Regarding “large” vs “small data” & complexity.** Yes, our approach applies to “large data” too. We apologize that this was not clear initially and thank R3 for pointing it out as well as the concern on scalability. Accordingly, we have revised Section 3.2 to make the full learning process and complexity clearer. In the initial draft, we highlighted why our approach was distinctive by enabling the analysis of small data as well as big data. In short, the learning complexity is between linear and quadratic, and is closer to linear in practice. Notably, although signals in Figure 4 seem “simple” (simplified for visualization and illustration purpose), the real task of discovering music theory is non-trivial: it requires handling higher-dimensional signals, and many signals (Paragraph 4, Section 4). Interestingly, the music app is actually an example of both “large data” and “small data”. On the one hand, the extremely rich music context requires the learning phase to handle numerous signals, i.e., numerous conditional probability distributions. On the other hand, for many conditional distributions, the event being conditioned on is rare, e.g., a “peculiar” multi-voice sequence that only appears one or two or even zero times in the data set (this is where abstraction comes in and plays a key role in making rare context common, e.g., although that particular multi-voice sequence is rare, its underlying chord progression might be widely seen in the data set).
>
> As R3 correctly pointed out, the construction phase, however, has an exponential complexity in the worst case. But the worst case is also a degenerate case, where a lattice reduces to a plain set with no ordering. In practice, the lattice is rich in structure (plenty of partial orders), and the exponential complexity is only a very coarse upper bound. This is empirically verified in domains like music and chemistry. Theoretically, we haven’t reached a result on the complexity of the full construction phase since it highly depends on the priors. As a new framework, many algorithms in the construction phase are in a vanilla version. Instead of reinventing the wheels or customizing specialized algorithms, we borrowed many existing, generic algorithms in solving subproblems, e.g., orbit computation algorithms from computational group theory, as well as poset sorting and lattice completion algorithms. Hence, there is much room for improving the current vanilla version, which includes switching to new state-of-the-arts and deriving specialized algorithms. Besides algorithmic updates, our structure also has much room for parallel computing then getting support from hardware—like a GPU implementation in deep learning.
>
> **Regarding probability simplex.** R3 is absolutely correct. The definition of signal is indeed general. In this paper, signals are not restricted to a probability simplex, but are assumed to be non-negative and on a finite domain. This is hardly a restriction for most real-world digital signals, but allows signals (and rules) to be normalized to a probability simplex whenever needed, e.g., when computing a Shannon entropy. Further, Shannon entropy used in the paper is a design choice (for measuring randomness) but not a must-have component. One may design other alternative functionals measuring properties like sparsity or variance, where randomness may be approximated and the restriction on probability simplex may be lifted.
>
> **Regarding deep-learning incorporation.** We think R3 mentioned a very thoughtful suggestion. Currently, our model and deep-learning-based representation learning are used for different problems, with ours focused more on explainability rather than task-specific performances like classification or game playing. But it is very interesting to think about a deep-learning version of ILL or our ILL version of a deep learning method, e.g., info-lattice-GAN or lattice-VAE. One immediate thought is to use our framework as a pre-processing or post-interpretation module for a deep learning model to achieve improved task performance and efficiency, as well as controllability and interpretability. Another thought may use our approach to analyze attention matrices (as signals) learned from a Transformer-based NLP model like BERT or GPT. We now include additional discussion of this in the last paragraph of Section 5.
>
> We thank the reviewer for these comments as well as suggestions on exploring the exciting intersection of lattice and deep learning, which we are preparing to pursue.

---

### Official Review · AnonReviewer4 · 2020-11-03
**Technical contents can not match motivation**

**Rating:** 4
**Confidence:** 4

**Review:**

This paper has addressed a very ambitious goal about explainability and generalizability from “small data" by generalizing the information lattice defined by Shannon. The topic of this paper is very significant but there are a few questions that I concern:

1. The paper has tried to answer some well-known challenging problems in machine learning such as explainability and generalizability from a very different perspective. However, the authors simply introduce some kind of framework but not provide a persuasive analysis or theoretical/empirical results to show it addressing the problems in the introduction. In fact, I do not find a theorem or commonly recognized experiment comparison in this paper. Thus I can not evaluate the significance of the technical contents.

2. The paper has used very complicated notations such as up/down arrows to show their results. However, is it really necessary? The tractability of the resulted problems (1) and the relaxed version (4) should be seriously concerned, not only providing a certain explanation or heuristic. Meanwhile, I recommend the authors to use simple and explicit enough formulations to show their framework so that we can know the tractability at the first glance, such as the convexity/nonconvexity, continuity/discontinuity, etc.

3. The experiments may be of interest to domain experts. However, it is not very attractive to the general audience? If the structure is really useful, can it be used to generate new music or find new chemistry laws? As the authors concern about generalizability in the introduction, I believe such reports are necessary. Meanwhile, if the framework is really useful, can it be used in the commonly accepted tasks and be compared with state of the art methods such as the deep learning approach mentioned in the introduction?

---

> ### Author Response · Authors · 2020-11-25
> **Response to R4**
>
> We are grateful for R4’s appreciation on the significance of the topic and perspective of our paper.
> 1. We regret that our initial draft was weak in persuasive analyses/results. We have now strengthened the experimental part. Due to the difference in the problems we solve, the experiments and their associated evaluation metrics are different from those in classical ML tasks.
>
>  &nbsp;
>
>  The music and chemistry examples in Section 4 are the empirical illustrations on how our model addresses the core problem of *explaining a signal*. First via two simplified examples (Figure 4, simplified for visualization purpose), we analyzed model outputs by describing how to read the outputs and how they correspond to known knowledge. Even from these simplified cases, we showed that our model recovered (in explicit forms) almost all static rules on two-voice counterpoint, and discovered an unexpected mutual interpretation between music and chemistry.
>
>  &nbsp;
>
>  Besides the simplified examples, we have now better presented two more evaluations in the real case of learning to write chorales. In Figure 5, we clearly list the three evaluation metrics we measure:
>
>  (a) how much known can be reproduced;
>
>  (b) how interpretable via human-subject studies;
>
>  (c) how much new can be discovered.
>
>  Notably, (a) is commonly used in knowledge discovery; (b) is commonly used in the HCI and XAI community; (c) is often lacking in many other similar works, but we have it.
>
> &nbsp;
>
> 2. We apologize for the non-standard notation of up/down arrows notating lifting/projection operators, which we agree caused additional burden in reading the results. Yet, this is a decision we have to make due to necessity. The two operators are essential in this paper, as they are the force behind lattice learning and decomposition-and-synthesis (Figure 1). The two arrows are the only notation we invent for this paper due to their importance and symbolic simplicity, which further clarifies other formulae. We agree that it takes time to become familiar with the two symbols, but hope that, in the end, this will make other parts much easier to read. Hence, we have strengthened the description of these two arrows, and simplified Figure 2 to help get the gist without referring to the precise math statements.
>
>  &nbsp;
>
>  We also thank R4 for pointing out the tractability issues which helps us better clarify them in the revision. Problem (1) is intractable but best for conceptual understanding. It was relaxed into (4) then into a sequence of (6)s which is tractable. We add three things to Section 3.2 to make the full learning procedure clearer. First, we explicitly state that the entire learning involves solving a sequence of *combinatorial* optimizations on a directed acyclic graph via BFS, i.e., Sequence (7). So, not continuous, not gradient-based. Second, we add the total complexity (between linear and quadratic) and point out that the richness of the partial-order structure determines whether it is closer to linear or quadratic (in practice it is far from quadratic). Finally, we add Figure 3B to outline a full run on the optimization sequence, which particularly shows how the search space evolves along the sequence. So, both the search procedure and the search complexity can be readily read off from this new figure.
>
> &nbsp;
>
> 3. Yes, the model can find new rules (e.g. unresolved tritones, music “second derivatives”, and “figured soprano” in the last paragraph of Section 4), but is not designed to generate new music (cf. “music theory is necessary but not sufficient for good music” in Section 5). The model can also find new interpretations of existing rules (cf. “cast chemistry laws as music theory and vice versa” in Section 4).
>
>  &nbsp;
>
>  For comparison to classical ML tasks and deep learning, R5 pointed out two important issues that require more effort. In the revision, we have better explained how we attempted to address them as a starting point and what remains challenging in the long run. At the current stage, our approach and deep learning solve different problems. Accordingly, our evaluation metric has followed the conventions in knowledge discovery and XAI tasks, which are also different from evaluating classical ML tasks. It remains a continuing effort to develop a systematic way and benchmark to compare models designed for AGI and XAI purposes. Yet, this is also commonly known as more challenging than classical benchmarks e.g., the ones used for classification or game playing where a winning criterion can be clearly computed then compared. We hope the added elaborations on the two studies in Section 4 can serve a good starting point for this long-term effort. Second, we are also trying to incorporate our approach into models like deep learning. Section 5 now has both points more clearly stated. Thank you for these important suggestions, and recommending future exploration at the intersection of lattice and deep learning, which we are preparing to pursue.

---

### Official Review · AnonReviewer5 · 2020-11-06
**A highly mathematical proposal for finding simple rules that explain a given signal**

**Rating:** 4
**Confidence:** 4

**Review:**

The authors propose an approach to explain a given signal $\xi$ (i.e., some function of interest, such as a 2D image, or a probability distribution) by learning simple "rules" that can accurately reconstruct it. They demonstrate their approach on a music dataset and a chemistry dataset.

I like the authors' introduction, problem statement, and the somewhat unusual viewpoint and the datasets. Despite this, I cannot recommend this paper for publication. The main issue is that, starting on page 4 and without clear justification, the authors introduce a nearly-impenetrable thicket of mathematical definitions and notation. I would be more accepting of this style if the approach and results absolutely necessitated it. However, it is not clear to me that this is actually the case --- as far as I can tell, what the authors propose to do is basically (1) generate a set of simple features (functions of the input space, created by composing various primitives and symmetries), (2) select a simple subset of these features that explain the target signal $\xi$ accurately. It seems to me that this kind of approach can be formulated without 90% of the machinery employed by the authors.  Even if it can't, the authors should start with a simple, understandable formulation of their approach, demonstrate the corresponding results, and -- if needed -- make it more complex in order to achieve better results.  Note also that, despite the high complexity, some basic elements of the approach, as are needed to understand the proposed objective function, are left undefined (for example, what does it mean to have the "Shannon entropy of a rule", $Ent(r)$, when $r$ is some arbitrary real-valued function?  What is the actual distance measure $\Delta$ used?)

Another major issue with this paper is that the author seem largely unaware of the closely related, and very well-established,  theories of induction coming from algorithmic information theory (AIT), as developed by Solomonoff, Chaitin, Rissanen (via minimum description length), and others. It seems to me that the proposed approach, of explaining a signal by finding simple rules that accurately reconstruct it, is basically trying to find a *compressed* version of the signal, i.e., a simple program for the signal, which is example the approach advocated by AIT. The relevant literature is too vast to mention, but one starting point could be
Chater and Vitanyi, Simplicity: a unifying principle in cognitive science?, TICS, 2003.

---

> ### Author Response · Authors · 2020-11-25
> **Response to R5 (regarding formulation)**
>
> We are delighted that our introduction, problem statement, as well as the non-traditional viewpoints were appreciated by R5. We also agree with R5 on how math should be used—we try to follow Occam's razor and believe in the power of simplicity. We apologize that the dense math caused difficulty in understanding, as our original intention was not to make things unapproachable but more precise while simplifying notation. We have smoothed out the math once more, and made sure all the necessary math is followed by explanation in plain language and/or examples. Below, we clarify possible confusion and why our formulations that rely on partitions and lattices rather than simple feature selection is essential for our task.
>
> We believe that summarizing our approach as a two-step feature selection process with (1) generating a set of simple features then (2) selecting a subset reflects an excellent “first model”. We now include that in the introduction as one of the two main intuitions behind our approach. Then we use that as the basis for developing our lattice-learning approach that improves on a simple feature selection approach in several ways. This is done by:
> 1. generating a **lattice** of **partitions**;
> 2. selecting a small subset via **structured search**.
>
> First, we argue that a **partition** (from a generated lattice) is more important than a **feature** for human learning. It captures the nature of human abstraction (as an equivalence relation). Although a feature function induces a partition (via preimages) as R5 suggests, there are many other distinct ways, which together can dramatically improve the likelihood that an explanation built upon them can be understood by human “students”, including:
> * symmetry-induced partitions (via orbits under groups of transformations);
> * partitions generated from other partitions (via meet/join).
>
> Note: although both features and transformations are functions of the input space, they play different roles (as highlighted in R5’s excellent suggested reference) in generating a partition: via a feature function, two data points are together iff they map to the same feature value; via a group of transformations, two data points are together iff they can be transformed from one to the other.
>
> Second, we argue that a **lattice** has more useful structure than a plain **set**. The structure is key, not only for producing more partitions, but also for **structured search**—which lies at the heart of lattice learning. To draw a conceptual connection, our partial-order-based search in a lattice is functionally parallel to a gradient-based search in the parameter space of a neural network: the lattice structure gives direction to the search by selecting candidates similar to the gradient in backpropagation.
>
> Another major difference between our **latticed-based rule selection** and **feature selection** in general is human-interpretability. We do not use arbitrary candidate features even if they are later proved effective for reconstruction, but are not interpretable. So, before selection, how to generate candidates is important, which motivates our approach of starting from human-like primitive priors (i.e., Core Knowledge) and combining them in a human-like way (i.e., composition of features, inclusion/exclusion of symmetries).
>
> ---
> **Two notational clarifications:**
>
> 1. We only deal with non-negative signals on finite domains in this paper (Paragraph 1, Section 2). So, all signals and rules are “normalizable”. Then, Ent(r) means the entropy of the normalized r, which agrees with the generalized definition of entropy, e.g., python’s entropy function. This restriction is not a limitation for two reasons: (a) there are ways to make the definition of a signal more general (Appendix B) but not necessary; (b) Shannon entropy here is a design choice (for measuring randomness) but not a must-have component. One may design other alternative functionals measuring properties like sparsity or variance, where randomness may be approximated and normalization is not needed.
>
> 2. $\Delta$ can be any distance function e.g., $\ell_p$ distance, or any divergence measure e.g., KL divergence (Paragraph 2, Section 3). All experiments in Figure 3 and 4 use $\ell_1$ distance, i.e., $\Delta (f,f') = || f-f' ||_1$. In the real music app, KL divergence $\Delta (f,f') = D_\text{KL}(f||f')$ is used in cases where we want to handle $\Delta(f,f')$ and $\Delta(f',f)$ asymmetrically, e.g., measuring a student mimicking Bach but not Bach mimicking a student.

---

> ### Author Response · Authors · 2020-11-25
> **Response to R5 (regarding AIT)**
>
> We thank R5 for pointing out AIT and we now feature it in explaining our approach. Indeed, our approach, as well as many other AI models (e.g., PCA, auto-encoder, compressed sensing, dictionary learning) and techniques (e.g., regularization, dropout, tree pruning), follow the same gist in AIT for advocating *simplicity*. All such models also share the same algorithmic viewpoint as *data compression*—using simple program/rules/code/latent space to reconstruct the original signal. Although different models all follow the same principle of simplicity, they differ by **how** and **for what purpose** simplicity is defined.
>
> **Our purpose for simplicity.** In many other models, simplicity is defined primarily for compression efficiency often in the form of task-specific performances, but not human-interpretability. We focus on interpretability. So, our model may be a lossy compression but requires both the compressed codes (i.e., rules) and the compression scheme (i.e., rule-learning process) interpretable. This shift of focus may be best seen in the example of AlphaGo, which clearly finds an optimal program to win against human masters. However, whether this “secret code” is human-interpretable (even by masters) is a different matter. In particular, can AlphaGo not only win games but also share with us its “secret code” and teach us how to play Go so well? This question motivated our approach for applications like learning music theory from music. And we evaluate our model NOT based on whether it “decompresses” into similar music, but based on whether it “compresses” music into rules similar to human-distilled rules as well as new rules.
>
> **Our definition of simplicity.** Different models use different forms and definitions of “simplicity”, based on the purposes mentioned above. AIT uses “program” and its shortest length; parametric models use parameters and regularization terms. In our case, we use *rules* as patterns on data abstractions and define *simple rules* if they are more deterministic or less random. Both definitions are used for formalizing our common-sense definition of a rule, and are consistent with many existing work (cf. references mentioned in Paragraph 3, Section 3).
>
> In short, as one of the core conclusions in AIT, “the length of a shortest program is nearly independent of the choice of a universal Turing machine”; however, the interpretability of such a program or such a Turing machine varies.
>
> In our initial draft, we cited works in the context of AGI and XAI, which are built on top of AIT but derived in the more specific context of mimicking human minds and human-like learning/generalization, e.g., Chollet’s “On the measure of intelligence” (https://arxiv.org/abs/1911.01547). We now explicitly feature AIT in our revision. We thank the reviewer for these comments and believe that responding to them has markedly improved the manuscript.

---

### Author Response · Authors · 2020-11-25
**Response to all**

We thank all the reviewers for their engaged and thoughtful reviews! Attending to their concerns has substantially improved the manuscript, and we summarize our revisions in the list below. We address points raised by each reviewer separately by following up the corresponding post.

---
**List of Revisions**

1. **Section 1, Paragraph 1:** add the three main empirical results (recovering known theory; discovering new theory; human-interpretability).
2. **Section 1, Paragraph 4:** add feature selection as one of the motivating intuitions.
3. **Section 1, last paragraph:** fix a typo and clarify the Tom-and-Jerry Example.
4. **Section 2, Figure 2:** simplify the figure to show important terms only and in particular, highlight the two most important terms: projection & lifting operators notated by up & down arrows.
5. **Section 3, opening paragraphs:** add motivation by and comparison to AIT; clarify notations.
6. **Section 3.1, Paragraph 2:** clean the math and clearly list the main steps.
7. **Section 3.2, Figure 3B:** add this figure and its corresponding caption to sketch an overview of the full learning path (e.g., the evolution of the search space).
8. **Section 3.2, Figure 3D:** add this figure and its corresponding caption to illustrate the shorthand notation of an ILL output and how to read such an output.
9. **Section 3.2, 2nd last paragraph:** add this paragraph to explain the full learning path summarized as Sequence (7) and its associated complexity. This paragraph aims to make both the learning process and the learning complexity clear.
10. **Section 3.2, last paragraph:** revise this paragraph to better explain how to read an ILL output represented by a two-line notation, making the following experimental results easy to understand.
11. **Section 4, last two paragraphs and Figure 5:** clearly state the three evaluation metrics (recovering known theory; discovering new theory; human-interpretability) and the associated evaluation results.
12. **Section 5, Paragraph 2:** add more discussions on how our approach may be used with other AI models such as deep learning.

---

### Decision · Program_Chairs · 2021-01-07
**Final Decision**

**Decision:**

Reject

**Comment:**

This paper has been evaluated by four expert reviewers resulting in two rejections one marginal score and one acceptance recommendation. The authors provided rebuttals to the critiques, but they did not sway the reviewers' assessments. The prevailing impression is that the work is interesting but perhaps not yet mature nor organized enough to benefit the ICLR audience in its current form. There is also some vagueness left at the conceptual level, e.g. regarding the actual objectives -- some reviewers pointed out confusing entanglement of the concepts of simplicity and interpretability. Nonetheless, the paper presents an interesting work that will benefit from incorporating the constructive feedback received here